# A single cell atlas of frozen shoulder capsule identifies features associated with inflammatory fibrosis resolution

Frozen shoulder is a spontaneously self-resolving chronic inflammatory fibrotic human disease, which distinguishes the condition from most fibrotic diseases that are progressive and irreversible. Using single-cell analysis, we identify pro-inflammatory MERTK[low]CD48[+] macrophages and MERTK + LYVE1 + MRC1+ macrophages enriched for negative regulators of inflammation which co-exist in frozen shoulder capsule tissues. Micro-cultures of patient-derived cells identify integrin-mediated cell-matrix interactions between MERTK+ macrophages and pro-resolving DKK3+ and POSTN+ fibroblasts, suggesting that matrix remodelling plays a role in frozen shoulder resolution. Cross-tissue analysis reveals a shared gene expression cassette between shoulder capsule MERTK+ macrophages and a respective population enriched in synovial tissues of rheumatoid arthritis patients in disease remission, supporting the concept that MERTK+ macrophages mediate resolution of inflammation and fibrosis. Single-cell transcriptomic profiling and spatial analysis of human foetal shoulder tissues identify MERTK + LYVE1 + MRC1+ macrophages and DKK3+ and POSTN+ fibroblast populations analogous to those in frozen shoulder, suggesting that the template to resolve fibrosis is established during shoulder development. Crosstalk between MerTK+ macrophages and pro-resolving DKK3+ and POSTN+ fibroblasts could facilitate resolution of frozen shoulder, providing a basis for potential therapeutic resolution of persistent fibrotic diseases.

Intractable inflammatory fibrotic diseases affecting soft tissues including lung, liver, kidney and skin contribute to 45% of all-cause mortality[1]. Clinically effective therapies that promote resolution of fibrosis are currently lacking. Recent single-cell transcriptome sequencing (scRNA-seq) analyses identify multiple scar-associated cell lineages that populate fibrotic visceral tissues. For example, cirrhotic human livers show enrichment for pro-fibrotic pathways including TNFRSF12A, PDGFR and NOTCH signalling with the cirrhotic niche being comprised of TREM2 + CD9+ macrophages and ACKR1 + PLAVP+ endothelial cells[2]. In pulmonary fibrosis, a subset of alveolar macrophages from fibrotic lungs were found to express high levels of mediators such as *IL1RN, MMP9* and *SPP1* that are known drivers of

inflammatory fibrosis[3]. Tissue-resident fibroblasts are likewise major constituents of the inflammatory fibrotic niche. Single-cell atlases of fibrotic human and murine lungs identify proliferating pathogenic myofibroblast subsets that are enriched for *POSTN, CTHRC1, MFAP5* and *ACTA2*[3–5]. Interactions between tissue-resident fibroblasts and pro-inflammatory macrophages induce fibroblast activation and irreversible deposition of fibrotic extracellular matrix[1,6,7]. These studies advance understanding of the biological processes culminating in solid organ fibrosis; however, the cellular basis by which fibrosis could resolve is not known.

To understand how intransigent human fibrotic diseases might be therapeutically encouraged to resolve, we investigated frozen

e-mail: michael.ng@ndorms.ox.ac.uk; stephanie.dakin@ndorms.ox.ac.uk

shoulder as an example of a localised chronic inflammatory fibrotic disease affecting the shoulder joint capsule that is normally self-limiting over time[8]. Frozen shoulder affects 10% of the working population, causing pain and disability through severely restricted shoulder joint motion[9–11]. Although characterised by localised fibrosis, a role for systemic factors in the pathogenesis of frozen shoulder cannot be excluded as it has been associated with both diabetes[12] and Dupuytren's disease[13]. Tissue biopsies from frozen shoulder patients with advanced-stage disease express inflammation-resolving proteins and show reduced expression of NFκB target genes compared to patients with persistent shoulder tendon tears[14]. These findings suggest that in frozen shoulder, the target tissues exhibit a resolving trajectory during advanced disease, which mirrors the typical clinical picture of reduced pain, stiffness and ultimate resolution over time. The unique biology of this under-investigated condition has the potential to unlock the cellular and molecular basis by which soft tissue inflammation resolves, persists or leads to fibrosis. Given that frozen shoulder is normally self-limiting, we hypothesised that the shoulder capsule is 'primed to resolve', and that the cellular interactions between major tissue-resident cell types including fibroblasts and macrophages (Mφ) might provide a resolving fibrotic niche conducive to restoring tissue homeostasis. Understanding this cellular basis for successful resolution could help provide the precise biological cues required to push persistent fibrotic diseases towards a resolving trajectory.

As animal disease models do not accurately recapitulate frozen shoulder-affecting humans, we use scRNA-seq and multiparameter immunofluorescent histology to discover the cell types and micro-anatomical features that comprise the resolving fibrotic niche in frozen shoulder patient tissues. We identify a population of MERTK + Mφ residing in the capsule lining that are enriched for biological processes associated with the modulation of inflammation. Using micro-cultures of patient-derived cells, we demonstrate that MERTK + Mφ inhibit the inflammatory phenotype of capsular fibroblasts and reveal a cellular basis for resolution of frozen shoulder via integrin-mediated extracellular matrix remodelling. Cross-tissue analysis reveals that MERTK + Mφ in the shoulder capsule and in the knee synovium of rheumatoid arthritis (RA) patients share a common gene expression cassette. Finally, we demonstrate a possible embryonic origin for the cell types implicated in the resolution of frozen shoulder, suggesting that the cellular template for the resolution of fibrotic adult capsular disease is established during foetal development.

## Results

### Single-cell analysis of the resolving inflammatory fibrotic niche of the shoulder capsule

The shoulder capsule is comprised of a series of ligaments and a thin synovial layer encapsulating the glenohumeral joint. During frozen shoulder, the shoulder joint capsule becomes chronically inflamed and fibrotic, leading to significantly restricted range of motion. To understand the cellular basis by which inflammatory fibrosis associated with this condition ultimately resolves, we utilised well-phenotyped tissue biopsies from non-inflamed comparator and frozen shoulder patients to generate an atlas describing all the cell types comprising the adult shoulder capsule. Tissue biopsy samples were collected from the rotator interval of the shoulder capsule from male and female patients with a diagnosis of advanced stage frozen shoulder (duration of symptoms shown in Supplementary Data 1). Site and age-matched non-inflamed comparator tissue biopsies were collected from male and female patients undergoing shoulder arthroplasty. Histological examination of comparator capsule identified a clearly demarcated thin capsule lining and sparsely cellular underlying sub-lining region comprised of parallel-orientated collagen fibres (Fig. 1A). Conversely, tissue biopsies from frozen shoulder patients exhibited increased cellularity of both lining and sub-lining regions and increased

vascularity (Fig. 1A). After integration with Harmony (version 0.1.0)[15], Leiden clustering of 6818 cells isolated from 10 donor tissue biopsies (6 comparator, 4 frozen shoulder donors) revealed 5 major populations, each represented in cells from both comparator and diseased shoulder capsule donors (Fig. 1B and Figure S1A & B). Annotation of the stromal cells profiled from the adult shoulder capsule revealed that COL1A1 + COL3A1 + PDGFRB+ fibroblasts were the most abundant cell types followed by CD14 + CD68+ myeloid cells, CD3+T cells, PECAM1+ vascular endothelial cells and ACTA2+ mural cells (Fig. 1C & Figure S1C & D). Frozen shoulder patient tissues showed a trend towards increased proportions of lymphoid cells and PECAM+ vascular endothelial cells and a smaller increase in myeloid cells relative to comparator tissues. The proportion of fibroblasts was significantly reduced in frozen shoulder relative to comparator patient tissues (10% FDR; Fig. 1C). To support these observations at the protein level, we performed immunostaining to identify the major cell types comprising comparator and frozen shoulder patient tissues (Figure S1E). For quantification we segmented the resulting images into cell-associated 'tiles' by using the Voronoi algorithm to automatically place boundaries around the cell nuclei. On average, comparator shoulder capsule tissue samples comprised of 13.9% CD31+ vascular endothelial cells, 19.1% CD68+ macrophages and 67.0% DKK3+ fibroblasts. In comparison, frozen shoulder samples had increased relative proportions of CD31+ (22.6% vs 13.9%) and CD68+ (27.1% vs 19.1%) tiles, with a commensurate decrease in DKK3+ tile proportion (50.3% vs 67.0%). These proportional differences were not statistically significant (Figure S1E). In both conditions, CD31+ tiles were proportionally higher in the sub-lining region, whereas CD68+ tiles were proportionally higher in the lining region. Collectively, these data support our observations around the major cell types comprising comparator and frozen shoulder patient tissues, with the most abundant cell type being DKK3+ fibroblasts, followed by lining enriched CD68+ macrophages and sub-lining enriched CD31+ endothelial cells, in agreement with the scRNAseq data (Fig. 1B & C).

### Immune cell atlas of frozen and comparator human shoulder capsule

A separate analysis of the lymphoid cells revealed two distinct CD3+T cell clusters, NK cell and B cell clusters (Fig. 1D, Figure S2A & B, Supplementary Data 2). The CD4+ cluster expressed IL7R, CCR6, TIMP1, CD40LG and LTB; the CD8A/B+ cluster expressed GZMK and KLRB1 (Fig. 1E, Figure S2C). Within the lymphoid cell subset, the relative proportions of the T cell and NK cell clusters was similar between comparator and frozen shoulder patient tissues while the B cells showed a slight but non-credible decrease in frequency in the frozen shoulder samples (Fig. 1F). Geneset over-representation analysis using gene ontology (GO) Biological Processes (BP) in CD4+ cells revealed enrichment for 'T helper cell differentiation' and 'regulation of leukocyte proliferation'; CD8+ cells were enriched for leukocyte and natural killer cell-mediated cytotoxic processes (Figure S2D). Small clusters of NKG7 + NK cells expressing cytotoxic molecules including PFR1, GNLY, GZMB and CD79A B cells were also identified (Fig. 1E & Fig. S2D).

A separate analysis of the myeloid cells identified four clusters in adult shoulder capsule tissues (Fig. 1G, Figure S3A & B, Supplementary Data 2). They included two MERTK + Mφ clusters; a LYVE1[high] cluster expressing MRC1, MAF, COLEC12 and a second LYVE1[low] cluster expressing FCGR3A, ICAM, TNF, MMP9 (Fig. 1H). A MERTK[low]CD48[high] macrophage cluster expressing PTGS2, IL1RN and S100A8 and a small cluster of CD48+ monocyte-derived dendritic cells (CD1C, AREG, FCER1A) were also identified (Fig. 1H). The relative proportions of these myeloid clusters did not significantly differ between comparator and frozen shoulder patient tissues (Fig. 1I). To investigate possible changes in macrophage phenotype between frozen shoulder and comparator tissues, we performed pseudo-bulk based differential gene expression analyses Figure S3C–E (Supplementary

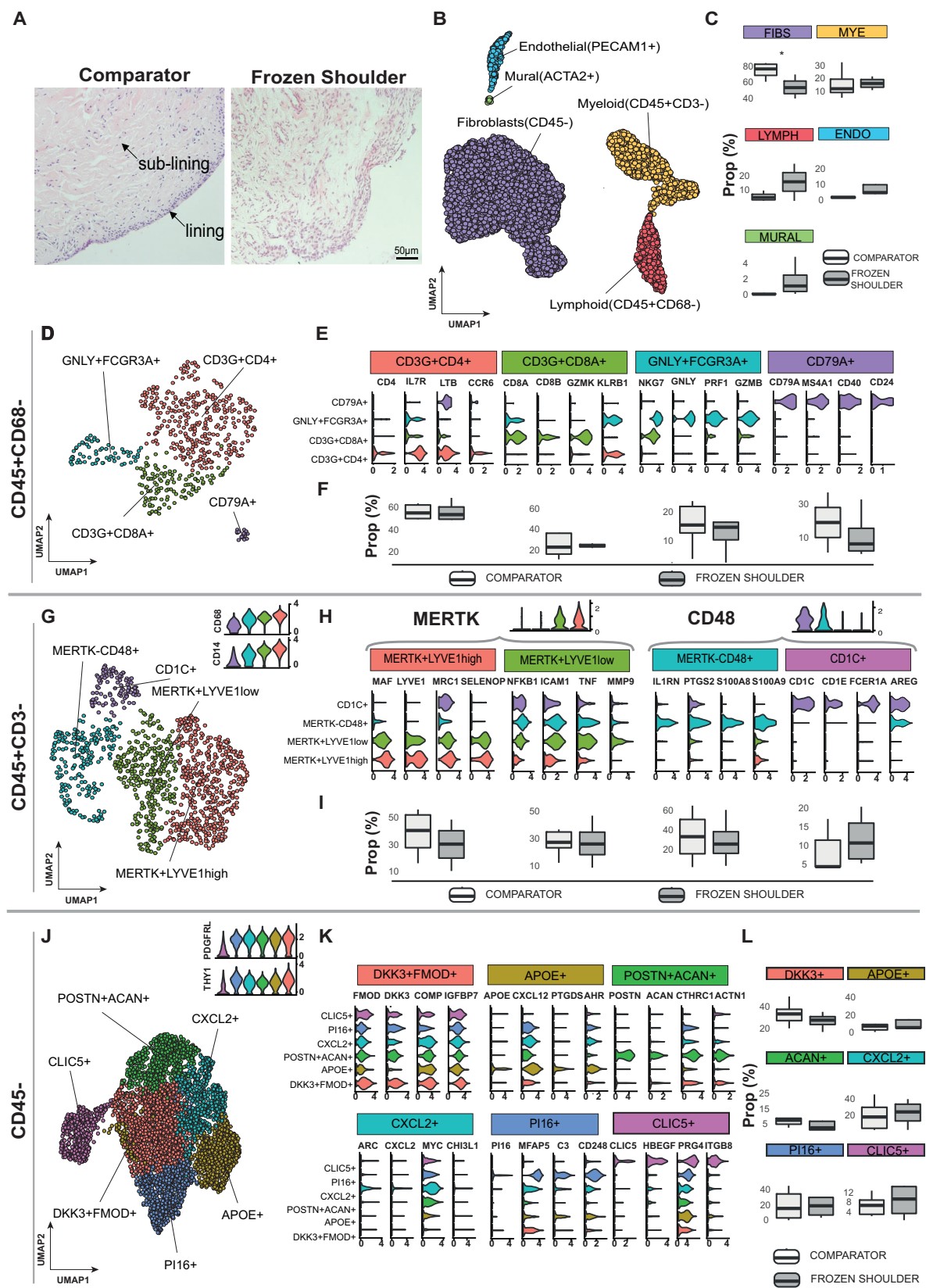

Data 3). Grouping of all myeloid cells revealed significantly elevated *SPP1* expression in comparator relative to frozen shoulder tissues (BH adjusted $P = 2.32 \times 10^{-9}$; DESeq2 analysis; Figure S3C, Supplementary Data 3). Within the macrophage clusters, we detected $n = 59$ genes in the *MERTK+LYVE1*low Mφ, $n = 92$ genes in the *MERTK*low*CD48*high Mφ and $n = 8$ genes in the *MERTK+LYVE1*high Mφ as differentially expressed between the frozen shoulder and comparator patient tissues. Of note, *MERTK+LYVE1*low Mφ isolated from frozen shoulder patient tissues had significantly higher expression of *CSF2RA, NFKB1, ITGAX* and *TNFRSF1B* (BH adjusted $p < 0.001$; DESeq2

**Fig. 1 | Cell types comprising the resolving inflammatory fibrotic niche.**
**A** Representative images showing Haematoxylin & Eosin staining of sections of comparator and frozen shoulder patient tissues. Distinct lining and sub-lining regions of the capsule are identified, frozen shoulder tissue sections show increased cellularity and vascularity relative to comparator tissues. Nuclear counterstain is violet, scale bar=50μm. scRNA-seq analysis of adult shoulder capsule from tissue biopsy samples collected from comparator (n = 6) and frozen shoulder (*n* = 4) patient donors. **B** UMAP shows the major cell types identified (resolution = 0.4) **C** The boxplots show the relative frequencies of the cell types in comparator and frozen shoulder patient tissues highlighting credible differences in the proportion of fibroblasts between these sample types (indicated by *). **D** The 4 identified lymphoid clusters (resolution = 0.2). **E** Violin plots of selected lymphoid cluster marker genes. **F** Relative frequencies of lymphoid clusters in comparator and frozen shoulder patient tissues. **G** The 4 identified myeloid clusters (resolution = 0.2). **H** Selected myeloid cluster marker genes. **I** Relative frequencies of myeloid clusters in comparator and frozen shoulder patient tissues. **J** The 6 identified fibroblast clusters (resolution = 0.3). **K** Selected fibroblast cluster marker genes. **L** The relative frequencies of the fibroblast clusters in comparator and frozen shoulder patient tissues. The credibility of differences in composition between the frozen shoulder and comparator samples was determined for each clusters **C**, **F**, **I** and **L** with scCODA[74] (10% FDR). Violin plots in **E**, **H**, **K** show log-normalized expression values of selected cluster marker genes. Only significant cluster marker genes are shown (two-sided Wilcoxon tests, BH adjusted *P* values < 0.05). Box plots in **C**, **F**, **I** and **L** show median as centre, lower and upper quartiles as box limits, and whiskers with a length of 1.5 IQR.

analysis; Figure S3D, Figure S4A, Supplementary Data 3). *MERTK+ LYVE1*^high^ Mφ isolated from frozen shoulder patient tissues showed little difference to those from comparators but had significantly lower expression of *SPP1* (BH adjusted p < 0.001; DESeq2 analysis; Figure S4A, Supplementary Data 3). *MERTK*^low^*CD48*^high^ Mφ isolated from frozen shoulder patient tissues had significantly higher expression of *CSF2RA*, *IL1A* and *IL1B* (BH adjusted *p* < 0.05; DESeq2 analysis; Figure S4A, Supplementary Data 3). The expression of selected pro-inflammatory and immunomodulatory genes in the myeloid cell clusters is shown in Figure S4A & B. We noted that the *MERTK+LYVE1*^high^Mφ cluster showed robust expression of immuno-modulatory genes including *CMKLR1*, *LYVE1*, *MRC1* and *TGFB* family genes in the frozen shoulder samples. Collectively, these findings demonstrate that comparator and frozen shoulder patient tissues are comprised of heterogeneous myeloid cell populations that have a modified transcriptional phenotype in frozen shoulder patients.

## Stromal cell atlas of frozen and comparator human shoulder capsule

Fibroblasts exist as distinct subsets with diverse roles in tissue homeostasis, inflammation and fibrosis[16–19]. They constitute the major cell type of the shoulder joint capsule, yet their precise phenotypes have not been described in this tissue. We therefore performed a targeted analysis of the *COL1A1 + COL3A1 + PDGFRB+* fibroblasts (Fig. 1J). After integration, six clusters were identified (Figure S5A & B). These included five THY1 + PDGFRL+ clusters: (i) "DKK3 + FMOD+" cells, (ii) "APOE +" cells expressing *CXCL12*, (iii) "POSTN + ACAN+" cells, (iv) "CXCL2+" cells expressing *ARC*, (v) "PI16+" sub-population expressing *MFAP5*. In addition, we found a *THY1*^low^*PDGFRL*^low^ "CLIC5+" cluster that also expressed *HBEGF* (Fig. 1K, Figure S5C, Supplementary Data 2). The relative proportions of fibroblast clusters did not significantly differ between comparator and frozen shoulder patient tissues (Fig. 1L). Geneset over-representation analysis revealed significant over-representations for the gene ontology (GO) biological processes related to collagen fibril and extracellular matrix organisation and 'response to mechanical stimulus' (DKK3 + FMOD+ and POSTN + ACAN + clusters), chemotaxis (APOE+ cluster), apoptotic processes (CXCL2+ cluster), complement activation, humoral immune response and angiogenesis (PI16+ cluster) and complement activation, TNF production and ion channel binding (CLIC5+ cluster) (Figure S5D). Pseudobulk-level differential expression analysis of fibroblasts in frozen shoulder relative to comparator patient tissues is shown in Figure S5E. Of note, *CLIC5*, *CSF1*, *ITGA8*, *SYNOP2* and *CXCL1* showed significantly higher expression in the fibroblasts from frozen shoulder patient tissues (BH adjusted *p* < 0.01; DESeq2 analysis; Figure S4C & D, Figure S5E and Supplementary Data 4). Taken together, these findings suggest that comparator and frozen shoulder patient tissues are comprised of distinct fibroblast sub-populations which are involved in shaping the extracellular environment and the regulation of local tissue inflammation.

## Spatial topography of the resolving inflammatory fibrotic niche

We next used immunofluorescence confocal microscopy to confirm the presence and to characterise the microanatomical niches of cell populations identified by single-cell analysis. Multiplex immunostaining using an extended panel of myeloid markers localised MerTK+Mφ subsets including LYVE1^high^ & LYVE1^low^Mφ to the capsule lining (Fig. 2A). A cassette of protein markers further validated these distinct clusters including LYVE1, CD163, MRC1 (LYVE1^high^ cluster) and FOLR2, ICAM1, CD83, FCGR3A (LYVE1^low^ cluster), located in the capsule lining region, this lining region was expanded in frozen shoulder patient tissues (Fig. 2A, Figure S3F & Fig. S3G). MerTK^low^CD48 + Mφ expressing PTGS2, IL1RA and S100A8 were identified in both lining and sub-lining regions and were abundant in frozen shoulder patient tissues (Fig. 2A, Figure S3F). Co-expression of Ki67 and CD163 support the proliferative phenotype of macrophages in frozen shoulder patient tissues (Figure S3H).

Quantitative analysis of immunopositive staining revealed an increased number of CD3+ cells in frozen shoulder relative to comparator patient tissues (Fig. 2B, *P* = 0.0006). CD3 + T cells resided adjacent to CD31+ vascular endothelium within the capsule sub-lining (Fig. 2B). Immunostaining confirmed the presence of CD4+, CD8+ and NK cells in frozen shoulder patient tissues, validating CD127 (CD4+ cluster), GZMB, GZMK, CD161 (CD8+ cluster) and CD56 + CD57+ (NK cell) populations (Fig. 2B).

Capsular fibroblasts also occupied distinct microanatomical niches. PDPN + PDGFRL+ fibroblasts populated lining and sub-lining regions, THY1+ (CD90 + ) fibroblasts predominated in the sub-lining adjacent to vascular endothelium (Figure S5F). CXCL12+, POSTN+ and MFAP5+ (PI16+ cluster) fibroblasts localised to sub-lining regions adjacent to blood vessels, DKK3 + FMOD+ and ARC+ (CXCL2+ cluster) fibroblast subsets were identified in both lining and sub-lining regions, CLIC5 + HBEGF+ fibroblasts mapped exclusively to the capsule lining (Fig. 2C). Mural cells (THY1 + ACTA2 + NOTCH3+) resided adjacent to vascular endothelium (Fig. 2C). Isotype control staining of capsular tissues is shown in Figure S6A–D. A summary of the cell types and microanatomy of the resolving inflammatory fibrotic niche is shown in Figure S6E.

Quantitative analysis of histological tissue sections identified increased cellularity across lining, sub-lining and combined regions of frozen shoulder patient tissues (*P* = 0.001, *P* = 0.0003 and *P* = 0.0003 respectively, Fig. 2D). Immunopositive staining for markers of tissue resident macrophages (CD68, *P* = 0.002), pro-inflammatory macrophages (CD48, *P* = 0.008), fibroblast activation markers (PDPN and CD90, *P* = 0.002 respectively) and matrix associated proteins (POSTN, CTHRC1, *P* = 0.008 respectively) were also increased in frozen shoulder relative to comparator patient tissues (Fig. 2D).

## Capsular MERTK^high^ macrophages have a modulatory phenotype

To identify candidate myeloid populations implicated in restoring homeostasis during advanced-stage frozen shoulder, we further

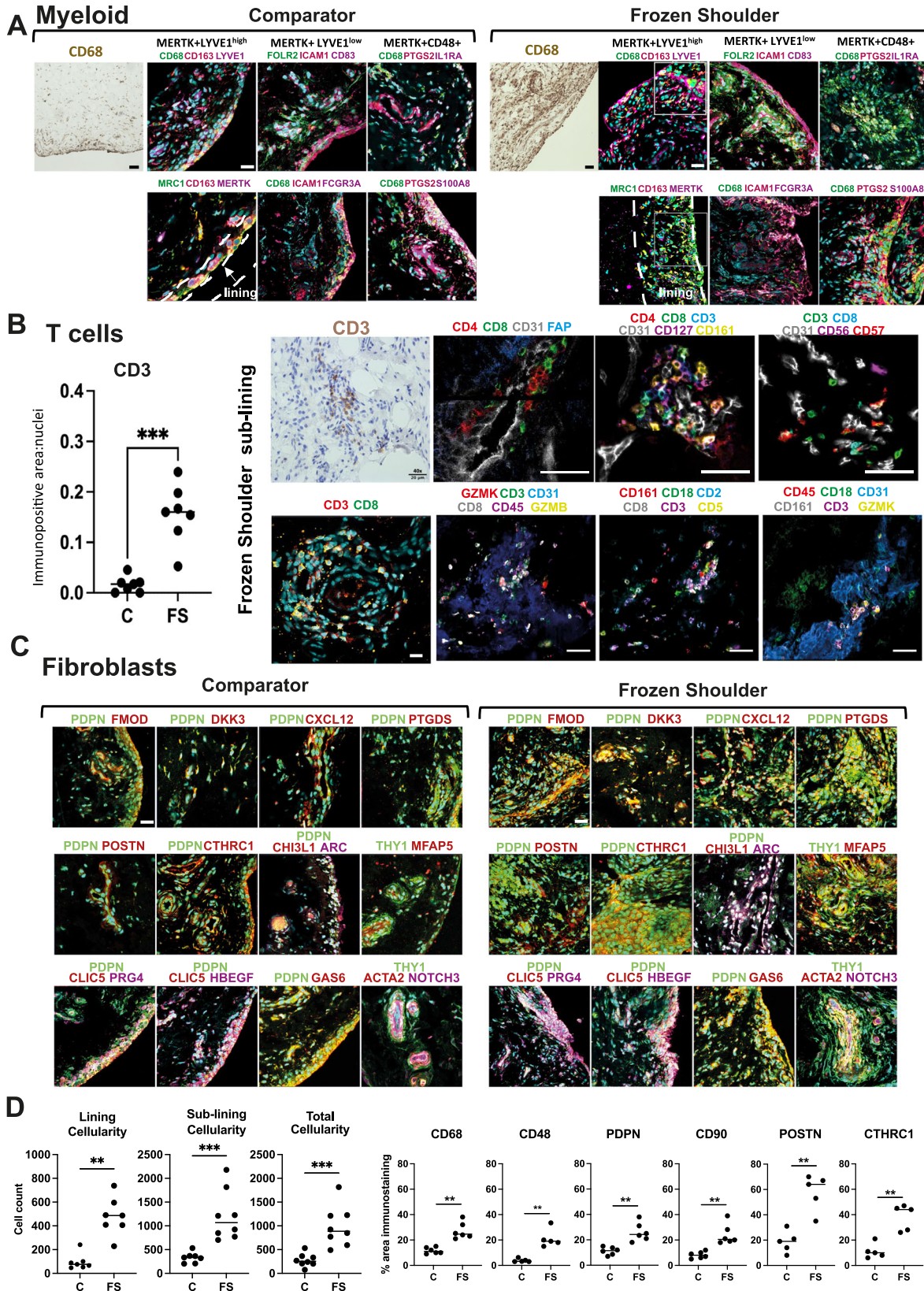

investigated the phenotype and biological processes active in these cells. MERTK+LYVE1^high macrophages expressed *MAF, COLEC12* and *SELENOP* (Fig. 3A). The marker genes for this cluster were significantly over-represented for biological processes concerned with the regulation of humoral immune responses, regulation of complement activation, acute inflammatory response, and receptor-mediated

endocytosis (Fig. 3B). Both MERTK+LYVE1^high and MERTK+LYVE1^low macrophage clusters expressed *CD163, FOLR2* and *MRC1* (Fig. 3C). The MERTK+LYVE1^low cluster exhibited a mixed phenotype also expressing *ICAM1, MMP9* and *TNF* (Fig. 3A & C). Biological processes significantly over-represented in the marker genes for this cluster included regulation of wound healing, cellular response to glucocorticoids and

**Fig. 2 | Spatial topography of the resolving fibrotic niche. A** Panel shows representative confocal images of immunostaining for CD68 and a wider panel of macrophage markers confirming the topographical niches of MerTK+LYVE1[high], MerTK+LYVE1[low] and MerTK[low]CD48+ macrophage subsets in sections of comparator and frozen shoulder patient tissues. MerTK + LYVE1[high]MRC1 + CD163+ and MerTK + LYVE1[low]FOLR2 + ICAM1 + CD83 + FCGR3A+ subsets predominate in the capsule lining; triple positive cells are highlighted with arrow heads (Figure S3G). MerTK[low]CD48 + PTGS2 + S100A8 + IL1RA+ macrophages occupy lining and sub-lining regions. Cyan represents POPO-1 nuclear counterstain, scale bar = 20 μm.
**B** Graph shows quantitative analysis of immunostaining for CD3 in comparator (C, *n* = 7 donors) and frozen shoulder (FS *n* = 7 donors) patient tissue sections, pooled from 3 independent experiments, statistically significant differences were calculated using a two-sided Mann-Whitney test (*P* = 0.0006). Panel shows representative ChipCytometry images of T cells in sections of frozen shoulder patient tissues, residing adjacent to vascular endothelium (CD31+). Panels show staining combinations for CD4 + T cells (CD127+), CD8+ T cells (CD161 + GZMK+) and NK cells (CD56 + GZMB+), nuclei counterstained cyan/blue, scale bar = 50 μm.
**C** Representative confocal images showing labelling for a cassette of fibroblast markers confirming the topographical niches of identified fibroblast sub-populations from Fig. 1J in sections of comparator and frozen shoulder patient

tissues. Sub-population markers include DKK3 + FMOD+, CXCL12 + PTGDS+, POSTN + CTHRC1+, ARC+CHI3L1+, MFAP5+ and CLIC5 + HBEGF + PRG4+. Cyan represents POPO-1 nuclear counterstain, scale bar = 20 μm. **D** Graphs show quantitative analysis (QA) of total cellularity and cellularity localised to lining and sub-lining regions. Data generated using tissue sections derived from a minimum of 7 comparator and 7 frozen shoulder donors, pooled from 3 independent experiments. Statistically significant differences were calculated using a two-sided Mann-Whitney test for lining cellularity (*P* = 0.001), sub-lining cellularity (*P* = 0.0003), total cellularity (*P* = 0.0003). QA analysis of macrophage markers in sections of tissue biopsies collected from comparator (C) and frozen shoulder (FS) patient donors including CD68 (*n* = 6 comparator and *n* = 6 frozen shoulder donors, *P* = 0.002) & CD48 (*n* = 5 comparator and *n* = 5 frozen shoulder donors, *P* = 0.008). QA for markers of fibroblast activation PDPN (*P* = 0.002) & CD90 (*P* = 0.002), (*n* = 6 comparator and *n* = 6 frozen shoulder donors for each marker). QA for matrix-associated markers POSTN (*P* = 0.008) and CTHRC1 (*P* = 0.008), (*n* = 6 comparator and *n* = 6 frozen shoulder donors for each marker). All immunostaining data were pooled from 4 independent experiments. Statistically significant differences were calculated using two-sided Mann-Whitney tests. Bars represent median values.
***P* < 0.001, **P* < 0.01.

leucocyte chemotaxis (Fig. 3B). Significant over-representations in MERTK[low]CD48+ macrophage cluster included those for biological processes associated with alarmins and pro-inflammatory mediators including leukocyte chemotaxis and migration, humoral immune response, and response to lipopolysaccharide (Fig. 3B). These findings support our hypothesis that MERTK+LYVE1[high] macrophages could restrain inflammatory fibrosis in frozen shoulder, providing a resolving fibrotic niche conducive to restoring homeostasis. Immunostaining of shoulder capsule tissue sections revealed that MerTK+ cells were predominately found in the capsule lining region and were more prevalent in frozen shoulder patient tissues (Fig. 3D & E, *P* = 0.005). We therefore investigated the cell types predicted to interact with MERTK+LYVE1[high] and MERTK+LYVE1[low] macrophage subsets that could restrain inflammatory fibrosis. MerTK, a member of the Tyro-Axl-MERTK (TAM) family of receptor tyrosine kinases is a macrophage receptor that mediates efferocytosis[20,21]. Engagement of MerTK by apoptotic cells, GAS6 or protein S (PROS1) triggers biochemically distinct responses mediating anti-inflammation and resolution[22]. *PROS1* and *GAS6* are ubiquitously expressed by fibroblast populations in the shoulder capsule, particularly APOE+ and CXCL2+ subsets respectively (Fig. 3F). Network Analysis Toolkit for Multicellular Interactions (NATMI) analysis[23] generated from comparator and frozen shoulder sub-populations (identified in Fig. 1) predicted specific interactions between MERTK+ macrophages and capsular fibroblasts, including PROS1 > MERTK (Sender cells>Target cells; APOE + >LYVE1[high] and APOE + >LYVE1[low]) and GAS6 > MERTK (APOE + >LYVE1[high], APOE + >LYVE1[low], CXCL2 + >LYVE1[high]) (Fig. 3G). Further investigation of possible interactions between these TAM ligands and their respective receptors identified GAS6 > AXL fibroblast autocrine circuits (Figure S7A) that were found to be most specific between APOE + > CLIC5+ and CXCL2 + > CLIC5+ clusters of the capsule lining (Figure S7B). Immunostaining confirmed the proximity of MerTK+ macrophages and GAS6+ and PROS1+ fibroblasts in frozen shoulder patient tissues (Fig. 3H). These findings further support the hypothesis that cellular interactions between MERTK+ macrophages and secreted ligands PROS1 and GAS6 from APOE+ & CXCL2+ fibroblasts could provide a resolving fibrotic niche conducive to restoring homeostasis in frozen shoulder.

## MerTK[high] and MerTK[low] macrophages induce divergent responses in capsular fibroblasts from frozen shoulder patients in coculture

To investigate how capsular macrophages and stromal cells might function to restore homeostasis during frozen shoulder, we explored

how the major macrophage subtypes present in the shoulder capsule could influence the phenotype of capsular fibroblasts. To model MerTK[high] and MerTK[low] macrophage populations identified in the shoulder capsule, we treated monocyte-derived macrophages (MDMs) with LPS (10 ng/ml)[24] or Dexamethasone (1uM) to induce MerTK[low] or MerTK[high] phenotypes respectively (validation of respective MerTK phenotypes are shown in Figure S7C & D). We directly co-incubated ex-vivo capsular fibroblasts from frozen shoulder patient tissues in the presence of either MerTK[low] or MerTK[high] MDMs for 48 hrs and performed bulk RNAseq on FACS sorted fibroblasts (Figure S7E). We compared the phenotypes of frozen shoulder capsular fibroblasts co-cultured with MerTK[low] or MerTK[high] MDMs relative to untreated fibroblasts (Figure S8A). In total, we identified *n* = 832 genes that showed significant variation in expression across the 3 groups comprising MerTK[low]-MDM co-cultured, MerTK[high]-MDM co-cultured capsular fibroblasts, or capsular fibroblasts in isolation (DESeq2, LRT, BH adjusted *p* < 0.01). These genes clustered into four groups with distinct patterns of expression across the three experimental conditions (Figure S8B), comprising of (i) a set of 345 genes that showed reduced expression in the MerTK[high]-MDM co-cultured fibroblasts relative to untreated and MerTK[low]-MDM co-cultured fibroblasts (group 1), (ii) a set of 311 genes which showed higher expression in MerTK[high]-MDM co-cultured fibroblasts (group 2), (iii) a set of 98 genes which showed higher expression in the MerTK[low]-MDM co-cultured fibroblasts (group 3) and (iv) a set of 78 genes which showed higher expression in the untreated fibroblasts (Figure S8B with selected examples shown in Fig. 4A). The set of genes downregulated by MerTK[high] MDMs in capsular fibroblasts relative to MerTK[low] MDM co-cultured and untreated fibroblasts (group 1) included pro-inflammatory genes (*TNFAIP6, PTGES, PTGS2, IL6*), chemotactic factors (*CXCL3*) and matrix metalloproteinases (*MMP1*). We noted that several of these genes, including *IL6*, *MMP1* and *CXCL3* showed higher expression in the fibroblasts co-cultured with MerTK[low]-MDMs relative to the untreated fibroblasts (Fig. 4A, Supplementary Data 5). Conversely, MerTK[high] MDMs induced a higher expression of genes (group 2) associated with maintenance and organization of the extracellular matrix (*NEXN, GSN, DPT, TIMP4, SPP1*), cell adhesion & migration (*VIT, EPHB6*), cell-matrix interactions (*THBS1*), immunoregulation (*FKBP5*) and cell proliferation (*TRNP1*) relative to MerTK[low]-MDM co-cultured and untreated capsular fibroblasts (Fig. 4A, Figure S8B, Supplementary Data 5), suggestive of a more homeostatic response. Co-incubation with MerTK[high] MDMs also down-regulated TGFβ responsive genes including *BMP2, BMP4, TGFBR1* and *TGFB2* (group 1, Supplementary Data 5). In support of these findings, the set of genes downregulated by co-incubating

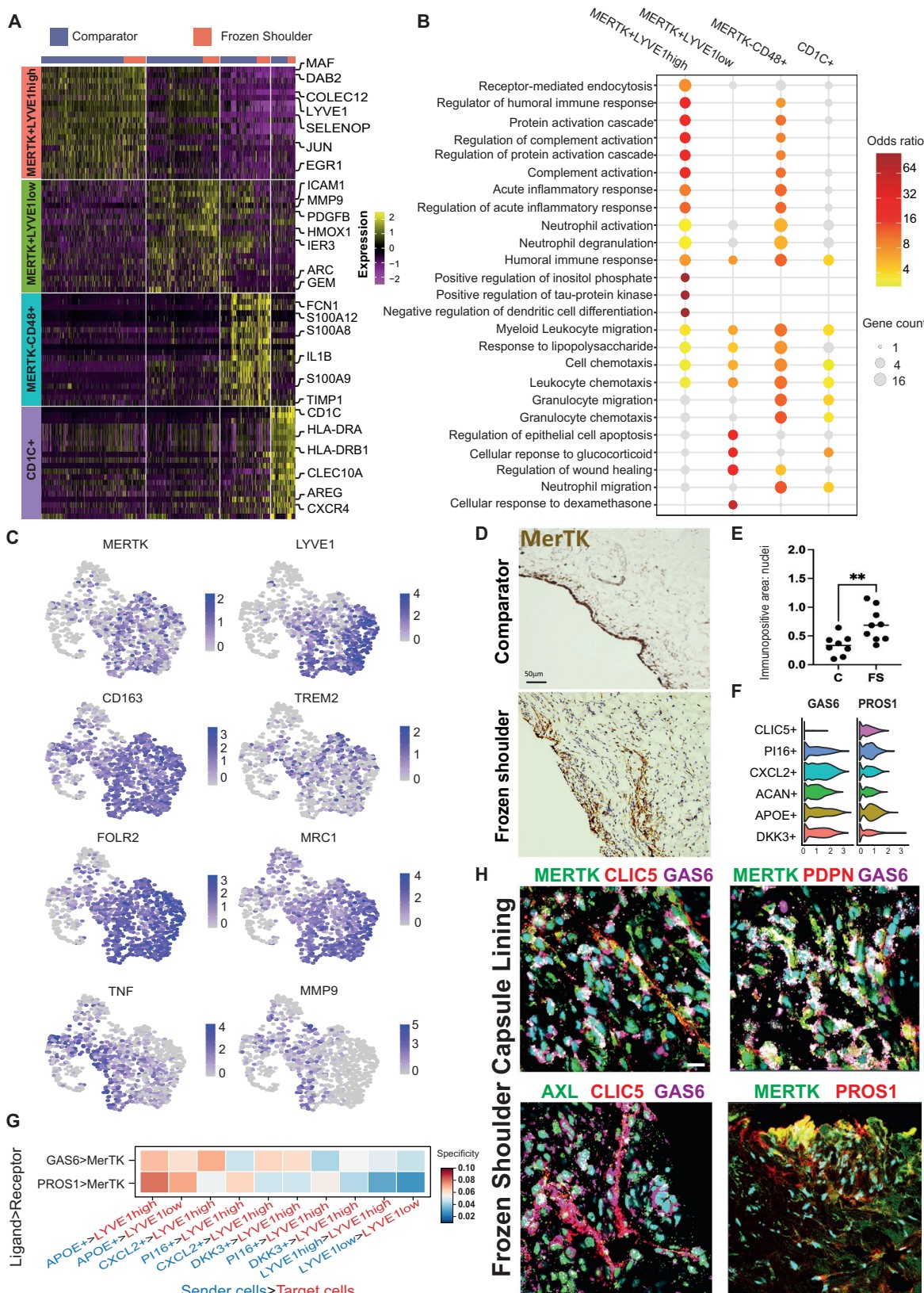

fibroblasts with MerTK^high MDMs was over-represented for biological processes including positive regulation of type 2 immune response, prostaglandin biosynthesis and transport, positive regulation of macrophage activation and epithelial to mesenchymal transition and negative regulation of tissue remodelling and apoptotic signalling (group 1, Fig. 4B, Supplementary Data 5). Conversely, genes up-

regulated by MerTK^high MDMs in capsular fibroblasts were over-represented for the biological processes 'negative regulation of leucocyte degranulation', 'negative regulation of IL-1β production' and those associated with cell-matrix adhesion, phagocytosis recognition, extracellular matrix assembly and disassembly, tissue remodelling and collagen metabolic processes (group 2, Fig. 4B, Supplementary Data 5).

**Fig. 3 | Capsular MERTK^high macrophages have a modulatory phenotype. A** The heatmap shows the top 20 marker genes for the capsular myeloid cell clusters (two-sided Wilcoxon test, BH adjusted $P < 1 \times 10^{-9}$). **B** Geneset over-representation analysis of gene ontology (GO) Biological Processes (BP) in the capsular myeloid clusters (one-sided Fisher tests, BH adjusted $P < 0.05$). **C** UMAPs show the expression of modulatory macrophage genes including *MERTK, LYVE1, CD163, TREM2, FOLR2* and *MRC1* in capsular macrophages, localising these markers to MERTK+ LYVE1^high and MERTK+LYVE1^low clusters. **D** Representative images of 3,3′-diamino-benzidine immunostaining (brown) for MERTK in sections from comparator and frozen shoulder patient tissues, staining is localised to the capsule lining region. Nuclear counterstain is haematoxylin, scale bar=50µm. **E** Graph shows quantitative analysis of MERTK+ cells in comparator (C, $n = 8$ donors) and frozen shoulder (FS,

$n = 9$ donors) patient tissues pooled from 2 independent experiments, bars show median values. Statistically significant differences were calculated using a two-sided Mann-Whitney test ($P = 0.0055$). **F** Violin plots show expression of MERTK ligands *GAS6* and *PROS1* in capsular fibroblast sub-populations. **G** Selected predicted ligand-receptor interactions between sender population (PROS1+ or GAS6+ fibroblasts) and receiver population (MERTK+LYVE1^high or MERTK+LYVE1^low macrophages) are shown. Predictions were generated from comparator ($n = 6$ donors) and frozen shoulder ($n = 4$ donors), sub-populations as in Fig. 1G (NATMI specificity score). **H** Representative immunofluorescence images of the lining region of frozen shoulder patient tissues, showing the topographical proximity of MERTK with associated ligands GAS6 and PROS1. Fibroblast markers include CLIC5, PDPN, AXL. Scale bar=20µm.

We next assessed if co-culturing MerTK^low or MerTK^high MDMs with capsular fibroblasts influenced the composition of fibroblast sub-populations in vitro. Deconvolution of bulk RNAseq data relative to fibroblast populations in Fig. 1J predicted that fibroblasts in the resting state (in the absence of MDMs) were comprised of DKK3 + FMOD+ (median 61%) and POSTN + ACAN+ (median 35%) sub-populations (Fig. 4C). Co-culture with MerTK^high MDMs did not induce a significant shift in the predicted composition. In contrast, co-culture with MerTK^low MDMs increased the predicted percentage of POSTN + ACAN+ (med 58%) cells and decreased the predicted percentage of DKK3 + FMOD+ (med 33%) cells. These findings suggest that under homeostatic and resolving inflammatory milieu, DKK3 + FMOD+ and POSTN + ACAN+ subsets are the predominant capsular fibroblasts sub-populations in vitro, and that MerTK^low MDMs can promote development of a POSTN + ACAN+ fibroblast phenotype at the expense of a DKK3 + FMOD+ phenotype. Immunocytochemistry confirmed expression of DKK3 and POSTN proteins in incubations of frozen shoulder capsular fibroblasts co-cultured with either MerTK^high or MerTK^low MDMs (Figure S7F, isotype controls shown in Figure S7G).

We applied NATMI cell-cell communication analysis to identify candidate signaling factors in MERTK+LYVE1^high macrophages for the induction of cellular and molecular pathways regulating extracellular matrix organization and structure in the DKK3 + FMOD+ or POSTN + ACAN+ fibroblast populations. Predicted interactions between these cell types were generated from comparator and frozen shoulder sub-populations (as per Fig. 1), identifying ligand-receptor pairs including F13A1 > ITGB1, CD14 > ITGB1, C1QB > LRP1, DCS2 > DGS2, EFNB1 > EPHB3, APOE > VLDLR, CCL8 > ACKR4, CCL13 > ACKR4 and LPL > SDC1 (Fig. 4D). Frozen shoulder patient tissues showed higher expression of predicted ligand-receptor pairs between the MERTK+LYVE1^high macrophages and DKK3 + FMOD+ or POSTN + ACAN+ fibroblast populations that included CXCL12 > ITGB3, FGF18 > FGFR1, EFEMP1 > EGFR and CXCL12 > ITGB1 (Fig. 4E). In contrast, we noted that comparator tissues showed a higher expression of predicted interactions involving *SPP1* expression by the MERTK+LYVE1^high macrophages (Fig. 4E). Protein-protein association analysis with IntAct[25] identified a possible link between the receptors highly expressed in frozen shoulder patient tissues and the transcription factor RUNX2 (Fig. 4F). Separately, single-cell gene-regulatory network analysis of POSTN + ACAN+ fibroblasts revealed a correlation between the expression of *RUNX2* and genes implicated in remodelling and organisation of the extracellular matrix including *CDH11, MMP14, MMP13* and *SPP1* (Fig. 4G). Relative to comparator tissues, frozen shoulder patient tissues showed a lower expression of predicted ligand-receptor pairs involving expression of the pro-inflammatory lymphotoxin-beta (LTB) cytokine by MERTK+LYVE1^high macrophages and LTB receptors by DKK3 + FMOD+ or POSTN + ACAN+ fibroblasts (including LTB > CD40, LTB > LTBR and LTB > TNFRSF1A, Figure S8C). Collectively these findings suggest that cellular interactions between MERTK+ macrophages and DKK3 + FMOD+ and POSTN + ACAN+ fibroblasts may play a role in the resolution of frozen shoulder by inducing integrin-mediated remodelling

of the fibrotic extracellular matrix and restraining the inflammatory phenotype of capsular fibroblasts.

## Cross-tissue comparison of MERTK+ macrophages in musculoskeletal tissues

Having identified a potential modulatory phenotype for MERTK+ macrophages in patient shoulder capsule tissues, we next compared capsular MERTK+ macrophages with a published dataset of MERTK+ synovial tissue macrophages (STM) isolated from rheumatoid arthritis (RA) patient cohorts[24]. MERTK + STM sub-populations were associated with RA disease remission, showed enrichment for negative regulators of inflammation and induced repair responses in synovial fibroblasts in vitro[24]. The transcriptome correlation distance between shoulder capsule and knee synovial MERTK+ and MERTK−CD48+ myeloid sub-populations is shown in Fig. 5A. MERTK+LYVE1^high macrophage clusters were common to both tissue types, the equivalent MERTK+ cluster in knee RA synovial tissues is annotated FOLR2^highLYVE1 +[24] (Fig. 5A). We applied a transfer-learning approach[26] to annotate the STM populations (Fig. 5B) with the cluster labels from our shoulder capsule dataset (Fig. 1G). The MERTK+LYVE1^high macrophage cluster in the shoulder capsule mapped to TREM2^low, TREM2^high, FOLR2^highLYVE1+ and FOLR2 + ID2 + STM sub-populations (Fig. 5C, reverse label transfer shown in Figure S8D).

Using transferred labels, we investigated the relative expression of myeloid target genes of interest including *MAF, LYVE1, MERTK, SELENOP, CD48* and *MRC1* in the shoulder capsule clusters and their corresponding (′) STMs subsets. Within the MERTK+LYVE1^high subsets, expression of genes implicated in restoring tissue homeostasis including *LYVE1, MRC1* and *MAF* was higher in shoulder capsule cells relative to the corresponding STMs subset (Fig. 5D, DESeq2 analysis, Wald Test, BH adjusted $P < 0.05$). To investigate whether these differences were representative of the broader MERTK+LYVE1^high phenotype we used a gene-cassette comprised of $n = 224$ shoulder capsule cluster marker genes for this population. A lower expression of this gene cassette was observed in the predicted knee synovium MERTK+LYVE1^high′ (Wald test, BH adjusted $P = 1.31 \times 10^{-199}$) and MERTK+LYVE1^low′ (BH adjusted $P = 5.75 \times 10^{-37}$) populations than was seen in the corresponding shoulder capsule clusters (Fig. 5E). We confirmed that this pattern was recapitulated after performing a reverse label transfer to identify a shoulder capsule FOLR2^highLYVE1+′ cluster analogue (Figure S8D, E). Inspection of the expression of the top 25 markers of MERTK+LYVE1^high shoulder capsule macrophages (Fig. 1) confirmed that while this capsule population and its predicted MERTK+LYVE1^high synovial counterpart had similar levels of expression of several marker genes including *C1QA, SELENOP, FOLR2* and *CD14*, the expression of others, including *LYVE1, MAF* and *MRC1* was less pronounced in the MERTK+LYVE1^high′ STM (Fig. 5F). Collectively, these findings show that while MERTK+ macrophages in the shoulder capsule and RA synovial patient tissues share a common gene expression cassette, MERTK+ macrophages in shoulder capsule patient tissues had a higher expression of genes implicated in restoring tissue homeostasis. To

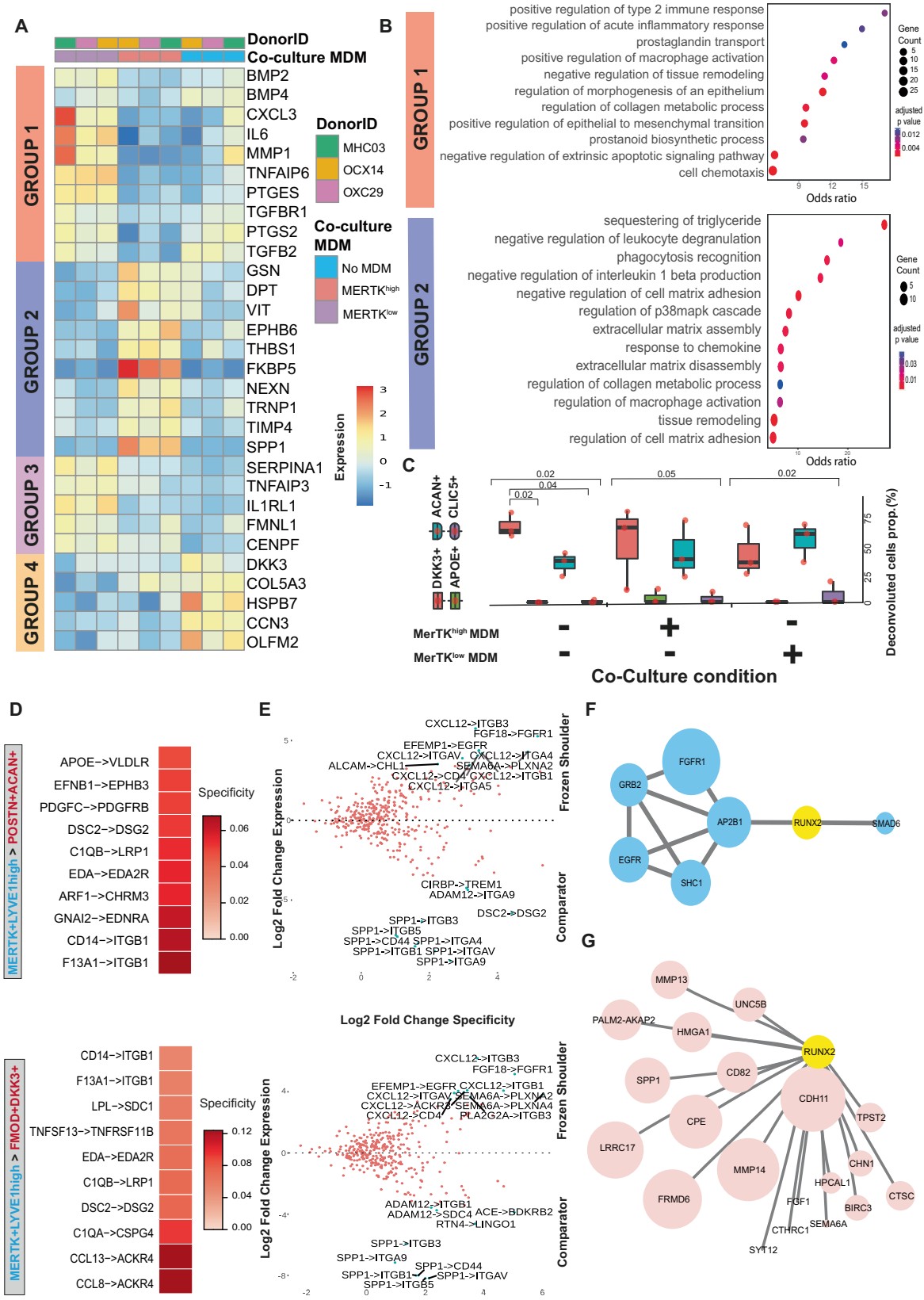

further understand the role of MERTK+ macrophages in frozen shoulder and RA, we compared gene expression changes in MERTK+ macrophages in these two diseases between patients and respective healthy controls. Overall, we observed a correlation of 0.21 (Wilcoxon test, $P = 2.42 \times 10^{-5}$) in the transcriptional changes within the MERTK+ macrophages between the two diseases (Figure S8F). To pin-point

genes that were regulated in different directions in disease in the two tissues we performed an interaction analysis. This identified 6 statistically significant genes that showed different directions of regulation in frozen shoulder vs RA. Relative to their healthy counterpart, we observed *AQP1, CLEC11A, CRIP2, EGR1* and *SEMA4A* being upregulated in diseased knee RA synovial tissues and downregulated in diseased

**Fig. 4 | MerTK^low and MerTK^high macrophages induce divergent responses in capsular fibroblasts from frozen shoulder patients. A** Heatmap showing selected genes that showed significant variance in expression between capsular fibroblasts from frozen shoulder patients co-incubated with MerTK^high, MerTK^low MDMs or fibroblasts in isolation (DESeq2, LRT test, BH adjusted $P < 0.05$). See also Supplementary Fig. 8B. **B** Dot plot shows GO biological processes over-represented in the sets of genes down-regulated in capsular fibroblasts by incubation with MerTK^high MDMs (group 1) or up-regulated by incubation with MerTK^high MDMs (group 2). **C** The boxplots show the predicted proportions of fibroblast subsets identified in the single-cell analysis (Fig. 1J) present in the untreated, MerTK^high-MDM co-cultured and MerTK^low-MDM co-cultured fibroblasts ($n = 3$ frozen shoulder patient donors in 2 independent experiments, bars show median value (median; centre, box limits; lower and upper quartile, whiskers; 1.5 IQR), deconvolution performed with MuSiC). **D** Selected predicted ligand-receptor interactions between MerTK+LYVE1+ myeloid and DKK3 + FMOD+ or POSTN + ACAN+ fibroblast sub-populations generated from differentially expressed genes from comparator ($n = 6$ donors) and frozen shoulder ($n = 4$ donors), sub-populations as in Fig. 1 (NATMI analysis). **E** The scatter plot shows the expression change of predicted ligand-receptor interactions between MerTK+LYVE1+ myeloid and DKK3 + FMOD+ or POSTN + ACAN+ fibroblast sub-populations in frozen shoulder relative to comparator patient tissues. **F** Protein-protein network association analysis of receptors (blue) highly expressed in frozen shoulder patient fibroblasts identified a candidate interaction with RUNX2 (analysis performed with IntAct). **G** Single-cell PySCENIC[66] gene regulatory network analysis of fibroblasts (Fig. 1J, all clusters) identified a connection between the expression of *RUNX2* and matrix associated genes including *CDH11, MMP14, MMP13* and *SPP1* (pink) in POSTN + ACAN+ fibroblasts.

---

frozen shoulder tissues (Figure S8G). Conversely, *BID* was upregulated in diseased frozen shoulder tissues and downregulated in diseased RA knee tissues relative to healthy comparators (Figure S8G, Supplementary Data 3). Despite these relatively minor differences, the overall similarity of the MERTK+ macrophage phenotype in frozen shoulder and RA supports the concept that this subset of macrophages may also be important for the resolution of frozen shoulder.

## The developing shoulder joint provides a cellular template for resolution

Ontological processes can be recapitulated as part of inflammatory disease pathology[27]. To explore if the cell types implicated in the resolution of inflammatory fibrosis in frozen shoulder are present during the development of the human foetal shoulder joint, we generated an atlas of human developmental shoulder tissues at 12, 15 and 17 post-conception weeks (pcw), (Supplementary Data 1). Following integration, clustering of 12,661 cells revealed 9 major populations, each represented in cells from 12, 15 and 17 pcw developmental stages (Fig. 6A, Figure S9A & Supplementary Data 6). The major cell types identified in our dataset include stromal fibroblasts, myeloid, progenitor and differentiated T cells and B cells, vascular endothelial cells, mural and cycling cells (Fig. 6A). Lymphoid populations were identified in foetal shoulder tissues from 15 and 17pcw developmental stages. These included CD3 + CD8+ T cells (GZMB + NKG7+), CD3 + CD4+ T cells (IL7R+) (Figure S9B), differentiated MS4A1 + B cells and three CD34+ B-cell progenitor clusters (CD38 + IGLC1+, MDM2 + RAG1+ and MKI67 + TOP2A+, Figure S9C). Immunostaining of developmental shoulder capsule tissues revealed CD19+ cells reside within CD31+ blood vessels, suggesting identified B cells as possible blood contaminants (Figure S9D). Clustering of foetal CD68 + CD14+ cells revealed 7 distinct populations including MERTK+LYVE1^high Mφ, MERTK + LYVE1^high TIMD4^high Mφ, MERTK^low S100A12 + Mφ, cycling MKI67 + TOP2A + Mφ, MYL4+ myeloid progenitors, CD1C+ monocyte-derived dendritic cells and CAECAM8+ neutrophils (Fig. 6B, Figure S9E, Supplementary Data 6).

Having identified interactions between MERTK+LYVE1^high macrophages and matrix associated DKK3 + FMOD+ and POSTN + ACAN+ fibroblasts induce matrix remodelling in cells from adult tissues, we investigated if correlates of these adult cell populations are present during foetal development. To do so, we transferred the cluster labels from the adult shoulder capsule myeloid clusters (as in Fig. 1G) onto foetal myeloid cells. The adult MERTK+LYVE1^high cluster mapped to TOP2A + MKI67+, MERTK + TIMD4+, MYL4+, MERTK + LYVE1+ and CD1C+ foetal myeloid sub-populations (Figure S9F). This analysis suggested that the MERTK+LYVE1^high macrophage phenotype is common to both adult and foetal shoulder tissues (Fig. 6B).

Analysis of foetal stromal cells identified fibroblast clusters including PI16 + MFAP5+, SCN7A + CCK+, WIF1 + NR4A2 + SCN7A + IRF1+, MFAP5 + IGFBP4+, TNMD + ITGA2, BGN + GEM+, APOE + FGF7+, MKI67 + TOP2A+ and HBEGF + CLU+ (lining layer) sub-populations,

COL2A1 + ACAN+ chondrocyte and TNMD + SCX+ tendon cell clusters (Fig. 6D, Figure S9G, Supplementary Data 6). Transfer of labels from adult fibroblast clusters (Fig. 1J) onto respective foetal fibroblasts suggested that similar DKK3 + FMOD+ and POSTN + ACAN+ fibroblast populations may be present in foetal shoulder tissues (Fig. 6E). The adult DKK3 + FMOD+ cluster maps to all foetal fibroblast sub-populations, the adult POSTN + ACAN+ cluster maps to all foetal fibroblast sub-populations except for SCN7A + CCK+ and HBEGF + CLU clusters (Fig. 6F). Feature plots showing expression of markers for major adult fibroblast populations (DKK3 + FMOD+ and POSTN + ACAN+) in foetal CD45– cells are shown in Figure S9H.

Finally, we spatially mapped myeloid and fibroblast populations implicated in the resolution of frozen shoulder onto foetal shoulder joint tissues to compare the respective topographical niches of these cells with adult tissues. Representative images of foetal shoulder joints are shown in Figure S10A & B, highlighting the anatomy of the shoulder joint capsule at 14 and 17pcw developmental stages. Immunostaining confirmed the presence of major cell types including fibroblasts (CD34, CD90), macrophages (CD68), T cells (CD4) and vascular endothelium (CD31, CD146) within the developing shoulder capsule (Figure S10C). CLU+ fibroblasts delineated the border between capsular fibroblasts and adjacent proximal humerus, NOV+ fibroblasts resided in the deeper sub-lining region (Figure S10 D & E). DKK3 + FMOD+ and POSTN+ fibroblasts were identified in the developing shoulder capsule (Fig. 6G & H). MERTK + LYVE1 + MRC1+ macrophages localised to the lining region of the developing shoulder capsule adjacent to CLU+ and GAS6+ fibroblasts (Fig. 6I & J, Figure S10F). Respective images for isotype control staining are shown in Figure S10G & H. Collectively these scRNAseq and spatial findings support an embryonic origin for MERTK+ macrophages and DKK3 + FMOD+ and POSTN+ fibroblasts in the shoulder capsule. This data supports the conclusion that the cell types implicated in the resolution of adult shoulder fibrosis are present during foetal development.

## Discussion

Frozen shoulder is a naturally occurring chronic inflammatory fibrotic human disease that is uniquely self-limiting over time. Analysis of well-phenotyped tissues collected from male and female patients during the advanced resolving disease stage provides an opportunity to identify the cell types and molecules implicated in fibrosis resolution. Using integrated scRNA-seq, spatial validation, cross-tissue comparison, and in vitro mechanistic experiments we provide new insights into the functional biology of this resolving inflammatory fibrotic niche. Systematic analysis of the cell types comprising the adult shoulder capsule during resting and diseased states identified distinct subsets of fibroblasts, T cells and macrophages occupying discrete microanatomical niches. Capsular fibroblasts share common features with those previously described in synovial tissues[16,18,19,24]. We identified two matrix-associated capsular fibroblast populations (DKK3 + FMOD+ and POSTN + ACAN+), two chemoattractant clusters

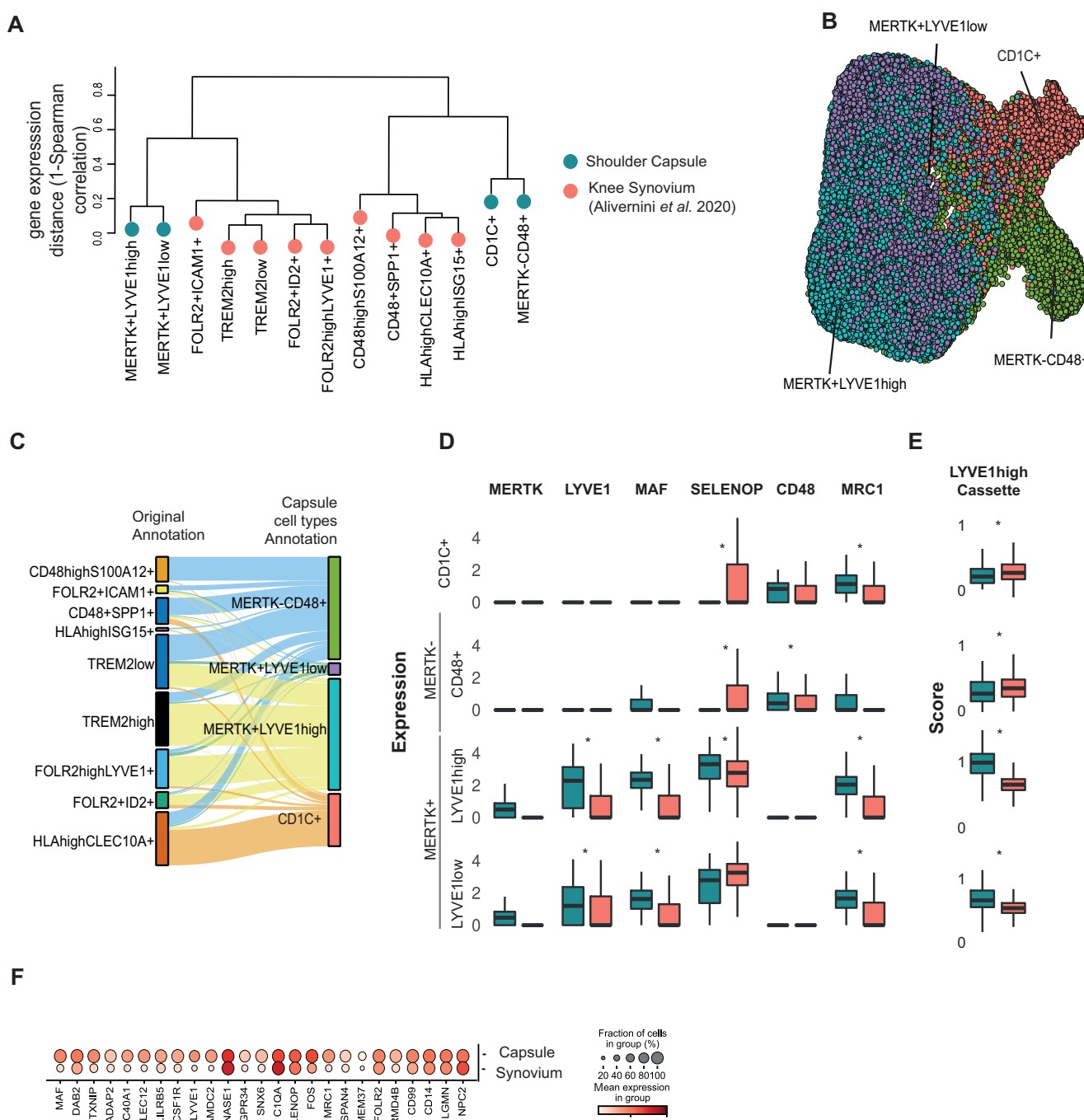

**Fig. 5 | Comparison of MERTK[high] macrophage clusters in shoulder capsule and knee RA synovial tissues. A** Dendrogram shows the transcriptomic Spearman correlation distance between human myeloid populations in comparator and frozen shoulder capsule tissues (blue) relative to knee RA synovial tissue macrophages (STMs, red) from Alivernini et al. (2020). In STMs, the MERTK + LYVE1+ cluster is annotated FOLR2[high]LYVE1+. **B** UMAP shows the Alivernini et al. (2020) STM cells with labels transferred from the myeloid clusters identified in adult shoulder capsule tissues (Fig. 1G) (scArches analysis) **C** The Sankey plot shows the mapping between the original STM sub-populations clusters identified by Alivernini et al. (2020) (left) and the transferred myeloid labels from the adult shoulder capsule tissues (MERTK−CD48+, MERTK+LYVE1[low], MERTK+LYVE1[high], CD1C+) (right). **D** The box plots show the normalized expression of myeloid genes including *MAF, LYVE1, MERTK, SELENOP, CD48* and *MRC1* in the shoulder capsule

myeloid clusters (blue, as per Fig. 1G, CD1C+ cluster *n* = 7; other cell types *n* = 8) and the corresponding STM subsets (red, as predicted by label transfer, all cell types *n* = 10). Bars show median value, *\*P* < 0.05 (Wald test). **E** Box plots (right) show cassette scores for marker genes of the MERTK+LYVE1[high] cluster in the shoulder capsule myeloid clusters (Fig. 1G, CD1C+ cluster *n* = 7; other cell types *n* = 8) and corresponding knee STM subsets (as predicted by label transfer, all cell types *n* = 10). Bars show median value, *\*P* < 0.05 (Wald test). Box plots in **D** and **E** show median as centre, lower and upper quartiles as box limits, and whiskers with the length of 1.5 IQR. **F** Dot plot shows average expression of top 25 MERTK+LYVE1[high] macrophage cluster marker genes in the shoulder capsule MERTK+LYVE1[high] macrophages relative to their corresponding cells in the Alivernini et al. (2020) STMs (as predicted by label transfer). Box plots in **D** and E show median as centre, lower and upper quartiles as box limits, and whiskers with the length of 1.5 IQR.

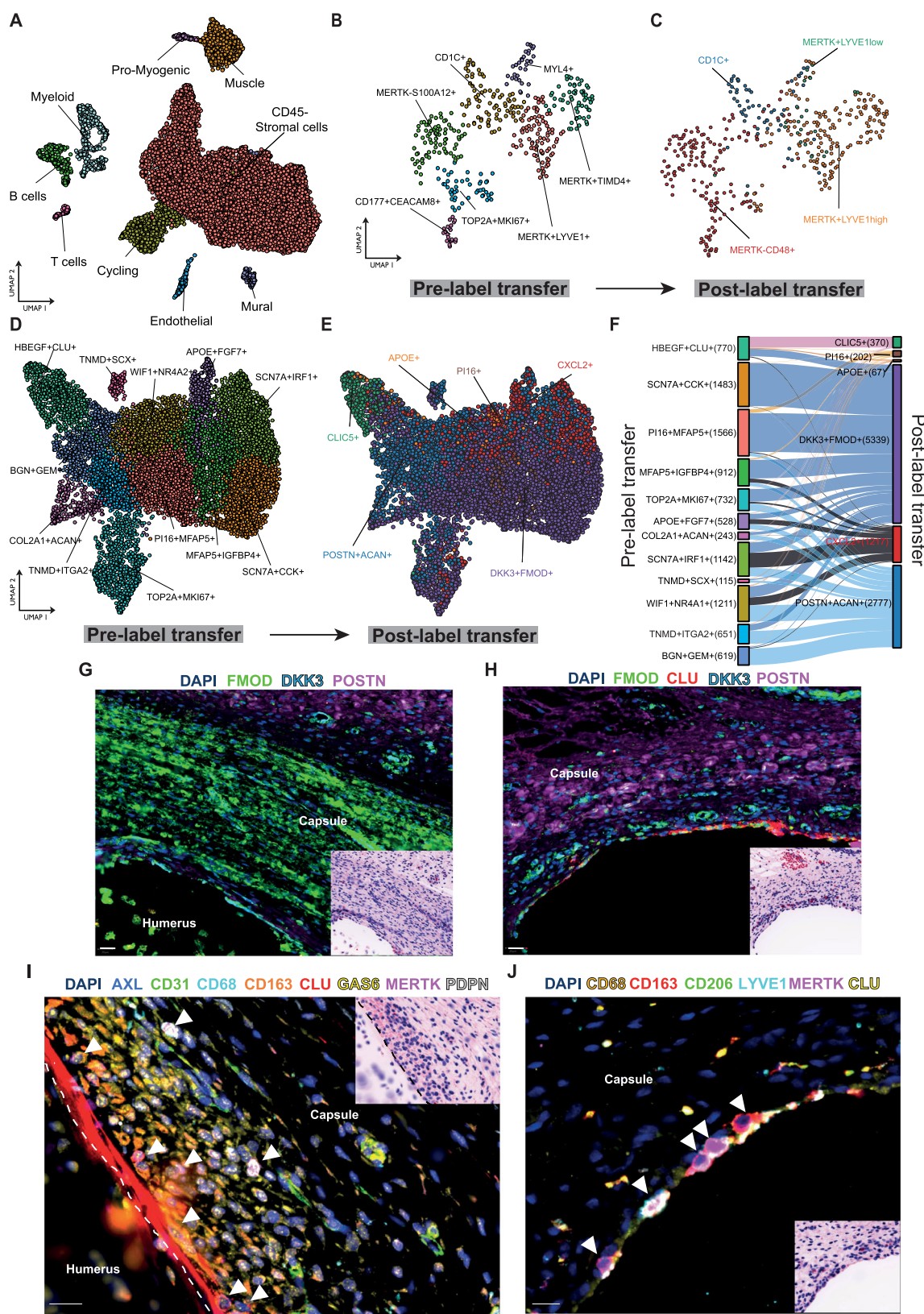

(CXCL12 + APOE+ and CXCL2 + ARC+), vascular interacting (PI16 + MFAP5+) and a lining layer cluster (CLIC5 + HBEGF+). Frozen shoulder patient tissues showed increased cellularity of lining and sub-lining regions relative to comparator patient tissues and highly expressed POSTN and CTHRC1 in the capsule sub-lining. Frozen shoulder patient

tissues were enriched for T cells which also resided in the capsule sub-lining. CD4 + CD127+ T cells expressed Th17-associated genes including *CCR6, RORA* and *AHR* as well as genes associated with a tissue repair phenotype[28–30]. This CD4+ subset was distinct from CD4+ clusters identified in patient RA synovial tissues[31]. CD8+ T cell and NK subsets

**Fig. 6 | The developing shoulder capsule informs a template for resolution.**
scRNAseq was performed on 12, 15 and 17 pcw developing human shoulder joint tissues. **A** UMAP shows the major cell types comprising the developing foetal shoulder joint (resolution = 0.9). **B** UMAP (resolution = 0.6) shows identified myeloid populations in the developing shoulder joint (12, 15 and 17 pcw). Sub-clustering of myeloid populations revealed 7 distinct clusters including MERTK + LYVE1+ and MERTK+TIMD4+ populations. **C** The plot shows the same foetal cell UMAP with labels transferred from the adult shoulder capsule myeloid cells (Fig. 1G) (scArches analysis). **D** UMAP (resolution = 0.7) showing CD45−COL1A1+ stromal cell clusters in the developing shoulder joint (12, 15 and 17 pcw). **E** The plot shows the same UMAP of developing shoulder stromal cells with labels transferred from the adult clusters (Fig. 1J) (scArches analysis). **F** The Sankey plot shows the mapping between the clusters identified in the developing shoulder stromal cells

(D) (left) and the transferred labels from the adult fibroblast populations (right). **G**–**J** Representative Cell DIVE and respective H&E stained images of histological sections of foetal shoulder joint at 17 post conception weeks (pcw) development stage. **G, H** Sections stained for a panel of markers to identify fibroblasts in the shoulder capsule during development (FMOD, DKK3, POSTN, CLU). DAPI nuclear counterstain is dark blue, scale bar = 20 μm. **I** High magnification image showing the cell types identified in foetal shoulder capsule including macrophages (CD68, CD163, MerTK) and fibroblasts (PDPN, CLU, AXL, GAS6) in the capsule lining region at 17 pcw. Scale bar = 20 μm. **J** High magnification image of the foetal shoulder capsule (17 pcw) showing immunostaining for macrophage markers MerTK, LYVE1, CD206 and CD163 in the capsule lining region. DAPI nuclear counterstain is dark blue, white arrows mark MerTK+ macrophages. Scale bar = 20 μm.

expressed granzyme, granulysin and perforin molecules akin to the SCT5 subset identified in knee synovial tissues from RA patients[31] and other inflamed human tissues[32]. Future studies are required to determine if these CD8+ cells contribute to resolution of frozen shoulder by facilitating the clearance of senescent cells.

MERTK^low^CD48+ macrophages from frozen shoulder patients highly expressed *IL1A* and *IL1B* although this inflammatory macrophage cluster showed reduced *SPP1* expression relative to tissues from C-19 and RA patients[33]. While it is plausible that this population of cells may play a targetable role in disease pathogenesis, future studies of patient tissues collected during earlier-stage frozen shoulder are essential to confirm if these inflammatory mediators are also active during early-stage disease. A population of MERTK + LYVE1 + MRC1+ macrophages resident in the capsule lining was enriched for biological processes concerned with modulating inflammation including regulation of humoral immune response, complement activation, inflammatory response, and receptor-mediated endocytosis. These findings support the hypothesis that MERTK+ macrophages might restrain inflammatory fibrosis, providing a resolving fibrotic niche conducive to restoring homeostasis in frozen shoulder, as observed at other sites of tissue repair[2,34–36].

Postulating that capsular MERTK+ macrophages aid resolution of frozen shoulder, we investigated how cellular crosstalk between MERTK+ macrophages and capsular fibroblasts from frozen shoulder patients could underpin fibrosis resolution. Co-incubation of capsular fibroblasts from frozen shoulder patients with MerTK^low^ MDMs induced an inflammatory fibroblast phenotype and biological pathways converging on chemotaxis and prostanoid biosynthesis. Conversely, co-incubation with MerTK^high^ MDMs negatively regulated leucocyte degranulation and IL-1β production pathways and upregulated genes and pathways associated with maintenance and organisation of the extracellular matrix and down-regulated TGFβ responsive genes including *BMP2, TGFBR1* and *TGFB2* relative to frozen shoulder capsular fibroblasts in isolation. Given that TGFβ induces the myCAF phenotype in cancer-associated fibroblasts[37], downregulation of TGFβ signalling could modulate the desmoplastic phenotype of capsular fibroblasts during frozen shoulder.

A recent study reported expansion of the DKK3+ synovial fibroblasts in treatment-refractory RA patients[38], however the precise function of this fibroblast population remains to be established[39,40]. Interestingly, DKK3 + FMOD+ fibroblasts in the shoulder capsule highly expressed inflammation-resolving receptor *CMKLR1*, which we previously identified in advanced-stage frozen shoulder patient tissues[14]. DKK3 + FMOD+ and POSTN + ACAN+ fibroblasts were the predominant populations isolated from patient shoulder capsule tissues utilised in micro-culture experiments with MerTK^high^ and MerTK^low^ macrophages. Predicted interactions between these cell populations revealed enrichment for ligand-receptor pairs including F13A1 > ITGB1, CD14 > ITGB1, C1QB > LRP1, DCS2 > DGS2, EFNB1 > EPHB3, APOE > VLDLR, CCL8 > ACKR4, CCL13 > ACKR4 and LPL > SDC1. Notably, frozen shoulder patient tissues showed enrichment for ligand-receptor

pairs including CXCL12 > ITGB3, FGF18 > FGFR1, EFEMP1 > EGFR and CXCL12 > ITGB1. ITGB1, ITGB3, EPHB3 and SDC1 are matrix-associated molecules. Fibroblast expression of integrins including ITGB1 is required to support tissue repair and remodelling[41]. Eph receptors and their ligands are essential to the development and function of neuro-muscular junctions and regulate actin cytoskeletal dynamics[42,43]. Syndecans including SDC1 mediate cytoskeletal organisation and remodelling of the extracellular matrix in cancer[44]. LRP1 promotes phagocytic activity, facilitating clearance of apoptotic cells[45] required for return towards tissue homeostasis. Interestingly, frozen shoulder patient tissues showed depletion for ligand-receptor pairs driving inflammatory processes including LTB > CD40, LTB > LTBR and LTB > TNFRSF1A. Protein-protein network association analysis of receptors highly expressed in frozen shoulder patient fibroblasts identified a possible role for the *RUNX2* transcription factor in regulating the expression of matrix-associated genes including *CDH11, MMP14, MMP13* in these cells. Collectively, these experiments using patient-derived cells inform interactions between MERTK+ macrophages and matrix-associated DKK3 + FMOD+ and POSTN + ACAN+ fibroblasts play a role in the resolution of frozen shoulder by inducing integrin mediated remodelling of the fibrotic extracellular matrix and restraining the inflammatory phenotype of capsular fibroblasts, suggesting a pro-resolving role for these cell populations in fibrosis resolution.

Synovial tissue macrophages (STM) from RA patients are comprised of distinct subsets, with patients in disease remission showing enrichment for MERTK+ clusters including TREM2^high^ and FOLR2^high^LYVE1+ macrophage sub-populations[24]. We compared MERTK+ macrophages in the adult shoulder capsule with knee RA STMs, identifying a common gene expression cassette between these tissue types. The overall similarity of the transcriptional signature of the adult shoulder capsule MERTK+ macrophages with remission associated STM populations supports a possible role for this myeloid population in also restoring homeostasis in frozen shoulder patient tissues. Intriguingly, we noted higher expression of the inflammation regulating genes *LYVE1, MRC1* and *MAF* in the frozen shoulder MERTK+ macrophages than was detected on their predicted STM counterparts. This suggests that these cells may have had a more potent resolving phenotype in the shoulder capsule samples in keeping with the unusually self-limiting nature of frozen shoulder.

Murine studies highlight perivascular MERTK+LYVE1^high^ macrophages maintain tissue homeostasis and prevent arterial stiffness and collagen deposition via MMP9-dependent proteolysis[46]. Depletion of LYVE1^high^MHCII^low^ monocyte-derived resident tissue macrophages in a murine lung fibrosis model exacerbated vessel permeability, immune cell infiltration, and collagen deposition, demonstrating their critical role in restraining inflammation and fibrosis[47]. In synovial tissues, MERTK + STM from remission RA patients were potent producers of inflammation-resolving lipids and induced repair responses in cultured synovial fibroblasts[24]. Collectively, these findings further support the ability of MERTK+ macrophages to restrain inflammation and support

homeostasis across multiple sub-tissular niches. Using microcultures of patient-derived cells, Alivernini et al. identified that MERTK + STM induced synovial repair responses in synovial fibroblasts from RA patients during disease remission[24]. Our functional experiments revealed that MerTK[high] MDMs restrained the inflammatory phenotype of frozen shoulder capsular fibroblasts and upregulated genes and pathways implicated in extracellular matrix organisation and remodelling which could also affect fibrosis resolution. Therefore, tissue-specific contexts could dictate the nature of this homeostatic response.

Having identified that crosstalk between MERTK+ macrophages and matrix associated fibroblasts underpin matrix remodelling in frozen shoulder, we investigated if the cell types implicated in the resolution of adult fibrotic capsular disease are present during foetal development. Single-cell analysis of human foetal shoulder tissues from 12–17 pcw developmental stages identified MERTK+ macrophage and DKK3+ and POSTN+ fibroblast cell populations that were predicted to correspond to equivalent populations identified in the adult tissues. In developmental tissues, MERTK + LYVE1 + MRC1+ macrophages localised to the shoulder capsule lining adjacent to GAS6+ fibroblasts as observed in adult tissues. The identification of MKI67 + TOP2A+ cycling myeloid cells supports the concept that myeloid cells divide and populate the soft tissues of the shoulder joint during foetal development, suggesting that MERTK+ macrophages are an embryonically seeded population. Adult frozen shoulder patient tissues also express Ki67, further supporting the concept of a self-sustaining macrophage population. The developmental origin of macrophages is well documented in visceral tissues including brain, heart, lung and liver[48,49] and the pool of LYVE1 + FOLR2+ macrophages are maintained through self-renewal with minimal monocyte input[50]. However, knowledge of the distinct immune populations occupying the soft tissues of the joint during human embryonic development is limited. Our findings demonstrate a possible embryonic origin for the cell types implicated in matrix remodelling of the fibrotic niche, suggesting that a template to resolve adult fibrotic disease might be laid down during foetal development. As ontological processes are frequently recapitulated in inflammatory diseases, we hypothesize that frozen shoulder could exemplify a recapitulation of ontogeny and during disease the re-activation of developmental cellular programmes could drive fibrosis resolution.

Our study builds upon previous work highlighting the importance of distinct phases of inflammation, resolution and remodelling in diseases of musculoskeletal soft tissues[51], generating new therapeutic targets that have the potential to exploit the cell-cell interactions we have observed in this uniquely self-limiting musculoskeletal condition. We have discovered a population of pro-inflammatory MERTK[low]CD48+ macrophages that may be involved in disease pathogenesis and provide functional evidence in support of a role for MERTK+ macrophages in disease resolution. We identify MERTK, LYVE1, MRC1 and DKK3+ and POSTN+ fibroblasts as specific targets for functional studies of fibrosis in murine models. Therapeutic enhancement of the functions of MerTK +Mφ and augmenting integrin mediated cell-matrix interactions between MERTK + Mφ and DKK3+ and POSTN+ fibroblasts to regulate inflammation and induce matrix remodelling could accelerate resolution of frozen shoulder and resolve persistent inflammatory fibrotic pathologies affecting other tissues.

## Methods

### Study approval
The University of Oxford Research Governance & Ethics Assurance approved the protocol for this study. Ethical approval for the use of adult tissues for this study was granted by the local research ethics committee (ICECAP study REC reference 18/SC0649, IRAS project ID 257757) and the Oxford Musculoskeletal Biobank (19/SC/0134). Full written informed consent according to the Declaration of Helsinki was

obtained from all patients. Foetal tissue samples were provided voluntarily with appropriate written informed consent, the Human Developmental Biology resource (HDBR) tissue bank operates under Research Ethics Committee approvals 18/NE/0290 (Newcastle upon Tyne) and 18/LO/0822 (London).

### Statistics & reproducibility
The objective of this study was to utilise a naturally occurring human disease model of self-limiting fibrosis (advanced-stage frozen shoulder) to understand the cellular and molecular basis underpinning successful fibrosis resolution. Adult tissue samples used for this study were collected from the rotator interval of the shoulder capsule from well-phenotyped comparator and diseased patient cohorts. Patients with frozen shoulder were included in the study if they had a diagnosis of frozen shoulder ≥12 months. Patients with concurrent rheumatoid arthritis, osteoarthritis and those receiving dexamethasone treatment 3 months prior to surgery were excluded from the study. Foetal shoulder tissues were collected from between 12-17 post conception week developmental stages. We performed single-cell RNA sequencing to identify the cell types comprising the shoulder capsule during developmental, adult comparator and adult diseased states. Functional experiments using micro-cultures of adult frozen shoulder patient-derived cells were studied to identify the cell types, molecules, genes and biological pathways underpinning fibrosis resolution in vitro. Statistical analysis and sample size justification were derived from previous studies (24, 49) that were sufficiently powered to gain insights from studying inflammatory and resolving processes in human musculoskeletal tissues. For single-cell RNAseq and bulk RNAseq data, investigators were not blinded to allocation during experiments and outcome assessment. Quantitative analysis of histological capsule tissues sections was performed by a single investigator who was blinded to the health status of the shoulder capsule tissue. For immunohistochemistry, H&E staining was first performed on sequential tissues to determine the region of interest in imaging, a minimum of 5 regions from each tissue slide were analysed. Representative images are shown for micrographs in Figs. 1A, 2A, 2C, 3D, 3H, 6G–J and supplementary figures S3F-H, S5F, S6A–D, S7F–G, S9D & S10A–H. Representative images are shown from a minimum of 3 independent experiments in each instance. The details of sample size and statistical tests employed in each case were provided in the figure captions. All $P$ values were corrected for multiple testing and the statistical testing method was indicated in the figure captions. We used the following convention to indicate significance with asterisks: not significant (ns) $(P > 0.1)$, * $(0.1 > P > 0.01)$, ** $(0.01 > P > 0.001)$, ***$(0.001 > P > 0.0001)$. No data were excluded from the analyses. The experiments were not randomized. Further information on research design is available in the Nature Reporting Summary linked to this article.

### Collection of adult shoulder capsule tissues
Comparator and frozen shoulder adult tissue biopsies were collected from the rotator interval of the shoulder (glenohumeral) joint capsule. Comparator tissues were collected from male and female patients undergoing elective shoulder stabilisation ($n = 8$ donors) or shoulder arthroplasty surgical procedures ($n = 12$ donors). Frozen shoulder tissues were collected from patients with advanced-stage disease (≥12 months symptom duration) undergoing surgical arthroscopic capsular release ($n = 15$ donors) (Supplementary Data 1). As frozen shoulder is a condition that affects males and females similarly, we ensured adequate representation of both sexes in comparator and frozen shoulder patient cohorts. All experiments included tissues or cells derived from both male and female donors.

### Processing of adult shoulder capsule tissues for scRNAseq
To identify the cell types comprising the resolving fibrotic niche, patient shoulder capsule tissue biopsies ($n = 6$ comparator, $n = 4$

frozen shoulder donors) were disaggregated and digested in DMEM F12 media containing 4 mg/mL Worthington Collagenase II (Lorne) and 1 mg/mL DNase (Lorne) at 37 °C for 90 mins with gentle agitation. The cell suspension was filtered and transferred to DMEM containing 10% foetal bovine serum (Gibco).

### Processing of foetal shoulder joint tissues for scRNAseq

Upper limbs ($n = 6$) from male and female foetuses were collected by the Human Developmental Biology Resource (HDBR) team following medical or surgical termination of pregnancy (Supplementary Data 1). Developmental stage was determined by anthropometric parameters and number of somites present. Collected tissue samples were stored in L-15 medium at 4 °C during shipment, time between collection and subsequent processing was <5 h. Soft tissues of the shoulder joint were isolated, disaggregated and digested using Liberase TL (Roche) 0.1 mg/ml diluted in DMEM (Gibco) over 2 h at 37 °C. Liberated cells were collected every 20 minutes during digestion, followed by replacement of digestion media. The isolated cell suspension was filtered and transferred to DMEM containing 10% foetal bovine serum (Gibco).

### scRNAseq of adult and developmental shoulder tissues

Single cell suspension samples were stained with 7-AAD dye (Bio-Legend 420404) for live/dead sorting on a SONY SH800 cell sorter. The density of cell suspensions was determined with the Bio-Rad TC20 Automated Cell Counter before loading 20,000 – 30,000 viable cells per sample onto a Next GEM Chip G and running on the 10x Chromium Controller. Single cell gene expression libraries were prepared using the 10x Chromium Next GEM Single Cell 3' Reagent Kits v3.1 following manufacturer user guide (CG000204). The final gene expression libraries were sequenced on the Illumina NovaSeq6000 platform (v1.5 chemistry, 28 bp Read1 and >91 bp Read2) to a minimum depth of 50,000 reads per cell.

### Processing and DAB immunostaining of adult shoulder capsule tissues

Shoulder capsule tissue biopsies collected from the rotator interval of comparator and frozen shoulder patients were immersed in 10% buffered formalin for a minimum of 48hrs. After fixation, samples were processed using a Leica ASP300S tissue processor and embedded in paraffin wax. Tissues were sectioned at 6µm using a rotary RM2135 microtome (Leica Microsystems Ltd) onto adhesive glass slides and baked at 60 °C for 30 mins and 37 °C for 60 mins. We used immunohistochemistry to validate key sub-population markers for macrophages and fibroblasts identified in Fig. 1. Comparator and frozen shoulder capsular tissue sections were obtained through deparaffinization and target retrieval steps (high pH, heat-mediated antigen retrieval) using an automated PT Link (Dako). Antibody staining was performed using the EnVision FLEX visualization system with an Autostainer Link 48 (Dako). Antibody binding was visualized using FLEX 3,3′-diaminobenzidine (DAB) substrate working solution and hematoxylin counterstain (Dako). Details of antibodies and their working dilutions are shown in Supplementary Table 1. For negative controls, the primary antibody was substituted for universal isotype control antibodies: cocktail of mouse immunoglobulin G (IgG1), IgG2a, IgG2b, IgG3, and IgM (Dako) and rabbit immunoglobulin fraction of serum from nonimmunized rabbits, solid-phase absorbed (Dako). Isotype control images are shown in Figure S6A-D. After staining, slides were taken through graded industrial methylated spirit and xylene, mounted in DPX mounting medium (Fischer Scientific) and imaged on a Brightfield microscope (Olympus). Images were acquired on a MOTIC slide scanner (Leica Biosystems) by a single blinded investigator. Image analysis was conducted using FIJI (National Institutes of Health version 2.3.0) using previously validated methodology[52]. For each sample, immunopositive staining was normalized to the number of hematoxylin-counterstained nuclei within the field of view. Data are presented as % area immunostaining. Statistical analyses were performed with GraphPad Prism, version 9.2.0 (GraphPad Software). Statistically significant differences in immunopositive staining were calculated using the Pairwise Mann Whitney U test. $P < 0.05$ was considered statistically significant.

### Validation & spatial mapping of distinct macrophage & fibroblast subsets in adult capsular tissues

We performed multiplex immunostaining to validate the phenotype of identified macrophage and fibroblast clusters and to identify the topographical niches these cells occupy in comparator and frozen shoulder adult capsular tissues using a previously validated protocol[51]. After antigen retrieval steps, tissues were blocked in 5% normal goat serum (Sigma) in phosphate-buffered saline (PBS) for 30 mins in a humid chamber at room temperature. Sections were incubated with the primary antibody cocktail diluted in 5% normal goat serum in PBS for 2.5 hrs at room temperature (primary antibodies used are listed in Supplementary Table 1A). Sections were washed with PBS–Tween 20 (PBST) and incubated in the secondary antibody cocktail, each diluted 1:200 in 5% normal equine serum (Sigma) in PBS for 2.5 hrs. The secondary antibodies were Alexa Fluor goat anti-mouse IgG2a or IgG2b or goat anti-rabbit IgG (Life Technologies) and goat anti-mouse IgG1 (Southern Biotech) (Supplementary Table 1B). After washing, sections were incubated in 2 mM POPO-1 nuclear counterstain (Life Technologies) diluted in PBS containing 0.05% saponin (Sigma) for 20 mins. Tissue autofluorescence was quenched with a solution of 0.1% Sudan Black B (Applichem) in 70% ethanol for 3 mins. Slides were mounted using fluorescent mounting medium (VectaShield), sealed, and stored at 4 °C until image acquisition. For negative controls, the primary antibody was substituted for universal isotype control antibodies: cocktail of mouse IgG1, IgG2a, IgG2b, IgG3, and IgM (Dako) and rabbit immunoglobulin fraction of serum from non-immunized rabbits, solid-phase absorbed (Dako) (Supplementary Table 1B). Isotype control images are shown in Fig S6C & D. Images were acquired on a Zeiss LSM 710 confocal microscope using a 40x oil immersion objective (numerical aperture, 0.95). The fluorophores POPO-1, FITC, Alexa Fluor 568, and Alexa Fluor 633 were excited using the 405, 488, 561, and 633 nm laser lines, respectively. To minimize bleed-through, all channels were acquired sequentially. Averaging was set to 2 and the pinhole was set to about 1 airy unit. Two-dimensional image reconstructions were created using ZEN Black software (Zeiss version 11.0.3.190).

### Validation of T cell subsets in frozen shoulder patient tissues

Multiplex ChipCytometry staining was used for immunophenotyping of T cell populations in frozen shoulder patient tissues using a previously validated protocol[53]. Cryosections were cut directly onto APES-coated coverslips (Sigma-Aldrich) and fixed immediately using freshly prepared 0.1 M phosphate-buffered 4% paraformaldehyde (Sigma-Aldrich) or Zellkraftwerk fixation buffer (Zellkraftwerk) for 10 mins at room temperature. After washing in PBS, sections on coverslips were assembled into tissue chips (ZellSafe Tissue – Chips; Zellkraftwerk). Tissue sections were blocked by incubating in 5% normal goat serum (Thermo Fischer Scientific) in PBS for 1 hr at room temperature. Immunostaining was performed at room temperature for 30 mins using 0.5 ml of Ab solution per chip. Antibody cocktails were diluted in PBS alone or PBS containing 2% normal goat serum. Markers were acquired in iterative rounds of photobleaching, staining, and imaging including CD45, CD3, CD4, CD8, GZMB, GZMK, CD127, CD40LG, CD18, CD161, CD2, CD5, FAP, CD31 (Supplementary Table 2). Images were downloaded from the proprietary software (Zellscanner App, version 19.08.2020, Zellkraftwerk Gmbh) as single marker 16-bit greyscale tif files. An outlier filter was applied to reduce signal noise (median, radius

0.5) and images were cropped, merged and colourised in FIJI (version 2.3.0) using previously validated methodology[54].

## Processing and immunostaining of developmental shoulder joint tissues

Foetal shoulders were fixed in 10% formalin for a minimum of 48hrs. Samples >10 post conceptional weeks were decalcified with 0.5 M EDTA solution for 15-30 days (depending upon developmental stage). Subsequently, samples were processed into paraffin using a tissue processor (Tissue TeK VIP 6 Processor, Sakura) and embedded in paraffin blocks in anteroposterior orientation. Tissues were sectioned at 6µm as per adult tissues. Multiplex immunostaining was performed to validate the phenotype and topographical niches of the major cell types comprising the shoulder joint capsule during development. Sections of foetal shoulder tissues were baked overnight at 60 °C. Slides were transferred to xylene for de-waxing followed by a series of rehydration steps in ethanol solutions (100%, 95%, 70%, 50%). Each step was repeated twice for 5 mins each, followed by 2 washing steps in 1x PBS. The slides were then permeabilised for 10 mins in 0.3% Triton X-100 and washed further in 1x PBS for 5 mins. Heat-induced Epitope Retrieval (HIER) was conducted with the NxGen decloaking chamber (Biocare Medical) with both Citrate (pH 6.0) and Tris (pH 9.0) antigen retrieval solutions. The decloaking chamber is programmed to incubate slides in Citrate solution (Vector Labs, H3300) for 20 mins reaching 110 °C and 6.1 PSI, which was maintained for 4 mins before cool down was initiated. Slides were transferred to a Tris solution for 20 mins followed by an additional 10 mins at room temperature (RT). Tissue slides were blocked with a 3% BSA (Merck, A7906) and 10% Donkey serum (Bio-Rad, C06SB) solution overnight at 4 °C. Slides were washed in 1xPBS for 10 mins and then stained with DAPI (Thermo, D3571) for 15 mins. Slides were washed in 1xPBS for 5 mins and cover-slipped with 75 µl of mounting media (50% glycerol – Sigma, G5516 and 4% propyl gallate – Sigma, 2370).

## Cell DIVE Imaging of adult & developmental shoulder capsule tissues

The GE Cell DIVE system was used to image adult and developmental FFPE slides using the ImageApp imaging software (version 1.0)[55,56]. An initial image using a 10x objective of the entire tissue allows for region of interest (ROI) selection. The background and innate autofluorescence of the tissue are captured during the initial 20x imaging round. This uses the FITC, Cy3, Cy5 channels as well as the DAPI signal to also create a virtual H&E image. Background imaging is used to subtract autofluorescence from all subsequent rounds of staining. Slides were de-coverslipped in 1xPBS prior to staining. Each staining round consisted of a master mix of 3 antibodies prepared in antibody diluent (3% BSA, 1xPBS). The initial round used primary antibodies which were incubated in the dark for 1 hr at room temperature followed by 3x washes in 1xPBS. Secondary antibodies raised in Donkey were then incubated for an additional hour at room temperature which were either conjugated to Alexa Fluor 488, 555 or 647. Each subsequent staining round used directly conjugated antibodies to either of these dyes. Fluorophores were bleached between each staining round using NaHCO$_3$ (0.1 M, pH 11.2. Sigma - S6297) and 3% H$_2$O$_2$ (Merck – 216763). Fresh bleaching solutions were prepared, and slides were bleached 3 times (15 minutes each) with a 1 minute 1xPBS wash in between bleaching rounds. Slides were re-stained for DAPI for 2 minutes and washed in 1xPBS for 5 minutes before imaging the dye-inactivated round as the new background round (for subsequent background subtraction). DAPI staining between imaging rounds assists in image registration and alignment. Slides were multiplexed with the next panel of three markers with iterative staining, bleaching and imaging. A negative control slide stained with isotype-matched antibodies was used at the same concentration as the corresponding primary antibodies and using the same exposure settings. Antibodies used for Cell DIVE immunostaining are listed in Supplementary Table 3. QuPath software (version 0.3) was used for image visualisation and the isotype controls were contrast-matched to samples stained with primary antibodies for comparison[57].

## Quantitative analysis of Cell DIVE staining in adult shoulder capsule tissues

To determine the relative proportions of cell types between comparator and frozen shoulder patient tissues, we performed Cell DIVE multiplex immunofluorescence staining for CD31, CD68 and DKK3 on comparator ($n = 5$ donors) and frozen shoulder ($n = 5$ donors) tissue sections. These markers were used to identify vascular endothelial cells, macrophages and fibroblasts, respectively. To quantify the result, we selected regions of interest from the staining and developed an analysis pipeline with custom code written in Matlab (version R2023a) and CellProfiler (version 4.2.5) to divide ROIs into tiles about cell nuclei (averaging 1058 tiles per ROI), then assign cell types to individual tiles from staining intensity. In brief, this pipeline tessellated ROIs using the Voronoi algorithm and DAPI nuclear staining to subdivide ROIs into multiple regions or "tiles". Each tile contains a single nucleus, and the tile borders are equidistant between neighbouring nuclei as a result of the Voronoi algorithm. The normalized highest intensity signal in each tile was calculated to assign each tile a state of either CD31+, CD68+ or DKK3 +, and the results shown in Figure S1E. This approach was used to minimise uncertainty from differences in cell size.

## Isolation of monocyte-derived macrophages from blood

Non-clinical blood cones were provided by the NHSBT and approved under REC number 11/H0771/7. PBMC were isolated using a density gradient (Histopaque, Sigma) and depleted from red blood cells using ACK lysis buffer (ThermoFisher Scientific). CD14+ CD16+ cells were isolated from peripheral blood mononuclear cells using EasySep™ Human Monocyte Enrichment Kit without CD16 Depletion (Stemcell) according to the manufacturer's protocol. Cells were differentiated into macrophages in complete media (RPMI, Gibco + FCS, Gibco +1% Penstrep, Sigma) containing 5% FCS and additional 100ng/mL M-CSF (PeproTech) with incubator conditions 37 °C/5% CO$_2$. On day 6, macrophages were harvested using cell lifters and plated in the appropriate plate at a density of 500,000 cells/well in a 6-well plate or 100,000 cells/well in a 96-well plate, in 3 mL and 200uL complete media, containing 3% FCS respectively and rested for 2 days. On day 8, cells were washed and stimulated with 10 ng/mL LPS (Invivogen) or 1uM Dexamethasone (Sigma) for 72 hrs. Doses of reagents used were adapted from a previously published protocol[24].

## Characterisation of Dexamethasone and LPS treated MDMs by flow cytometry

To characterise the blood cone derived macrophages after stimulating with Dexamethasone or LPS, protein expression was measured by flow cytometric analysis. For this experiment 4 different donors were used in independent experiments. Monocytes were isolated from blood cones and treated with M-CSF for 5 days as described above. After plating to 1×10$^6$, cells were rested for 2 days and stimulated with LPS or Dexamethasone to induce respective MerTK$^{low}$ or MerTK$^{high}$ phenotypes as above. Macrophages were incubated with Accutase (Invitrogen) for 10 mins then harvested with a cell lifter for flow cytometry. The cells were stained with antibodies against the surface markers MerTK, CD163, CD206, TREM2, CD14, LYVE-1 and CD48 (1:100 dilution, Supplementary Table 4A). In addition, a Live/dead stain (1:500) and FC block (1:100) was added to the mixture. After 20 mins of staining at 4 degrees, cells were centrifuged, and supernatant was discarded before cells were fixed using 4% PFA in PBS at RT for 20 mins. After centrifugation and discarding of the supernatant, cells were permeabilized for 15 mins with 1x BD Perm/Wash™ buffer and stained with antibody against CD68 for 20 mins at RT. Cells were washed in permeabilization

buffer and resuspended in FACS buffer (10% FBS in PBS) before running on an LSRII Flow Cytometer. Data was analysed using FlowJo V10. Data were expressed as geometric mean of staining intensity and were normalised within donors relative to control (minus Dexamethasone or LPS stimulation). Data (from $n = 4$ donors) were analysed in GraphPad Prism (version 9.2.0). Data are expressed as SEM, statistical significance was determined using an unpaired t-test, $P < 0.05$. Relative protein expression of MerTK, CD206 and CD48 across 4 donors is shown in Figure S7C, representative contour plots showing expression of MerTK, CD206 and CD163 are shown in Figure S7D.

## Coculture of MerTK[high] and MerTK[low] MDMs with primary capsular fibroblasts

Monocyte derived macrophages (MDMs, $n = 3$ donors) in 6-well plates were stimulated for 72 hrs with the previously described conditions to induce MerTK[high] or MerTK[low] populations. After washing with PBS, capsule-derived fibroblasts isolated from frozen shoulder patients ($n = 3$ donors) were added to MerTK[high] or MerTK[low] incubations at a density of 200,000 cells/well. Fibroblasts and MDMs were co-cultured for 48 hrs in complete media containing 3% FCS prior to harvest and sorting (Figure S7E). After 48 hrs, cells were incubated with Accutase for 10 mins and vigorously resuspended. Media was added and remaining adherent cells detached by scraping. A single cell suspension was obtained after putting cells through a 70 μm cell strainer and resuspended in Facs buffer containing 1% BSA 0.1 mg/ml DNase in PBS. Cells were stained with antibodies PDPN, CD90, CD45, CD14 (1:200 dilution, Supplementary Table S4B) to distinguish between fibroblasts and MDMs. Directly co-cultured MDMs and fibroblasts were sorted on a BD Aria III with Diva 8.01 software. Fibroblasts (PDPN + CD90 + CD31-) and MDMs (CD45 + CD14 + ) were sorted into RLT Lysis buffer (Qiagen). RNA from sorted capsular fibroblasts was extracted using the RNEasy microprep kit (Qiagen) according to the manufacturers protocol.

## Bulk RNA-seq of capsular fibroblasts co-cultured with MDMs

Total RNA was quantified using Quant-it RiboGreen RNA Assay Kit (Invitrogen), and the integrity assessed by the 4200 Tapestation system (Agilent). PolyA+ mRNA was enriched and purified from 100 ng high-quality total RNA (RIN > 9) using NEBNext Poly(A) mRNA Magnetic Isolation Module (E7490L). Generation of double stranded cDNA and library construction were performed using the NEBNext Ultra II Directional RNA Library Prep Kit for Illumina (E7760L), with custom adapters and barcode tags (dual indexing, based on https://doi.org/10.1186/1472-6750-13-104). Indexed libraries were multiplexed based on fluorescent-based quantification. The final size distribution of the multiplexed pool was determined using Tapestation and quantified by Qubit assay (Thermo Fisher), before sequencing on an Illumina NovaSeq6000 v1.5 in 150 bp paired-end mode. Data from the NovaSeq6000 were demultiplexed using bcl2fastq (Illumina) version 2.20.0. Cassette scores were calculated using addModuleScore function and the log normalised count of all MERTK[high] markers.

## Immunocytochemistry for markers of matrix-associated fibroblasts

Human monocyte-derived macrophages were seeded in chamber slides and stimulated for 72 hrs to drive either a MerTK[low] or MerTK[high] phenotype as described above, along with unstimulated 'M0' monocyte-derived macrophages as control. Following stimulation, primary frozen shoulder fibroblasts ($n = 3$ donors) were co-cultured with macrophages for 48 h prior to fixation with 4% PFA. Cells were permeabilized with 0.1% Triton-X, blocked with 5% normal goat serum in PBS + 0.1% Tween-20 (PBS-T), then stained with POSTN (1:200, abcam #ab79946) and DKK3 (1:400, Proteintech #66758)

overnight, followed by secondary antibodies at 1:200 (goat anti-mouse 633, goat anti-rabbit 568, Life Technologies). Nuclei were counterstained with POPO-1 (1:1000, ThermoFisher #P3580) in 0.05% saponin, and slides imaged using an inverted Zeiss LSM 710 confocal microscope with 10X objective. Respective isotype control staining is shown in Figure S7G.

## Computational analysis of single-cell RNA-sequencing data
### Analysis of single-cell RNA-seq data from frozen shoulder and comparator adult tissues.
Single cell RNA sequencing data was aligned to human genome using 10x Genomics Cell Ranger version 6.1. and 2020-A Reference annotations (GRCh38, GENCODE v32/Ensembl 98). Ambient background RNA was removed using CellBender (version 0.2.0; droplets included=10,000, fpr=0.01, epochs=150). A total of n = 13194 cells were identified for downstream analysis. Doublets were determined using Scrublet (version 0.2.3) at an expected doublet rate of 0.05[58]. The count matrix was transformed into a Seurat object for downstream analysis[59]. Additionally, we removed cells that had abnormally low or high gene counts ( < 300 or >6500 n_Feature) and high mitochondrial gene expression ( >10% mitochondrial genes, n = 5818 cells removed). Cell cycle scores were computed using curated gene sets[60] and the addModuleScore function provided in Seurat (version 4.0.4). Cell counts were log normalised, scaled, and n = 3000 highly variable genes (HVG) were selected using the VST method and used for principal component analysis (PCA, n = 50 PCs retained). We corrected the latent space using Harmony by the condition (i.e., comparator and frozen shoulder) of the samples (n = 15 components retained)[15]. Downstream analyses were performed using the Cellhub workflow (https://github.com/sansomlab/cellhub). Nearest neighbour graphs were computed with the HNSW algorithm (Euclidean distance, k = 20) and clustering performed with the Leiden algorithm. We applied additional filters to remove clusters of cells with low UMI (n_Count <10000) and gene count (n_Features <1000), and highly expressed hemoglobin markers (*HBA*, *HBB*, *HBG*) (n = 558 cells). The remaining clusters of high-quality cells (n = 6818 in total) were assigned to stromal, myeloid or lymphoid cell type "regions" based on singleR cell type predictions[61] (version 1.06) and expression of known cell-type specific marker genes (*CD34 PECAM−*, stromal; *CD14*, myeloid cells; *CD3*, lymphoid cells). The count matrices for each of the cell-type regions were extracted and separately re-analysed using the workflow described above. For the stromal cells we used n = 1500 HVG, n = 50 PCA components, n = 14 harmony components and a clustering resolution of 0.3. For the myeloid cells we used n = 400 HVG, n = 50 PCA components, n = 11 harmony components and a clustering resolution of 0.2. We removed a cluster of cells that consisted mainly doublets (score ≥ 0.4; n = 9 cells removed). For the lymphoid cells n = 1000 HVG, n = 50 PCA components, n = 11 harmony components and a clustering resolution of 0.2 was used. The markers of each cluster were identified using a Wilcox test (BH adjusted $P ≤ 0.05$).

### Pseudobulk-level differential expression analysis.
Within cell type differential expression analyses of the adult single cell dataset (comparator vs frozen shoulder) were performed at pseudo-bulk level. Pseudo-bulks were created by summing gene counts within cluster for each sample using Muscat. Differential expression analysis was performed using DESeq2[62] (version 1.26.0; ashr shrinkage, Wald test). Genes expressed (≥10 counts) in n ≥x samples where x was equal to the number of biological replicates (Supplementary Data 3 & 4) in any of the condition groups were retained for differential expression analysis. To identify genes that were regulated differently in frozen shoulder and RA relative to healthy controls we performed an interaction analysis. To do so, we created pseudobulk datasets as described above and employed an interaction term (condition:tissue) in the regression model (-condition + tissue). The log2 Fold changes were adjusted

using ashr shrinkage, and P-values were corrected for multiple testing (BH adjusted P ≤ 0.05). We used Wilcoxon test to compare the median log2 fold changes between tissue types, and calculated correlation coefficient using Spearman's correlation.

**Pathway analysis.** To further dissect the functionality of genes and gene sets that had been identified from the sequencing data, we performed pathway over-representation analysis using Gene Ontology database[63,64] and Canonical pathway databases (BioCarta, http://cgap.nci.nih.gov/Pathways/BioCarta_Pathways; KEGG, http://www.pathway.jp; Pathway Interaction Database, PID, http://pid.nci.nih.gov; Reactome, http://www.reactome.org and WikiPathways, https://www.wikipathways.org/) with gsFisher (https://github.com/sansomlab/gsfisher) and ClusterProfiler (version 4.2.2)[65]. For the analysis of the single cell sequencing data, we used marker genes (Wilcox, BH adjusted P ≤ 0.1) representative of each cluster and restricted the gene universe to genes that were detected in a minimum of 10% of cells. For the bulk sequencing data, we used the differentially expressed genes (DESeq2, BH adjusted P ≤ 0.05) and restricted the gene universe to include genes that were in the DEG analysis. Significant pathways were ranked and selected by their magnitude of the odds ratio, which was evaluated using Fisher Exact test and corrected for multiple testing using the Benjamini-Hochberg algorithm.

**Cell-cell interaction analysis.** We identified receptor-ligand interactions between cell clusters using Network Analysis Toolkit for Multicellular Interactions (NATMI)[23], we retained only fully validated interactions from literature and the constituent ligand and receptor genes that are expressed at a detectable level (>5% within cluster).

**Protein-protein and gene regulatory network analyses.** Protein association network analysis was performed using Intact[25] (release 242) and the resulting network graph was filtered to retain nodes with a maximum of two degrees of separation. Single-cell gene regulatory network analysis of the fibroblasts was performed using pySCENIC[66] (version 0.11.2) and the elicited biomolecular interaction networks and regulons were visualized using iRegulon[67] and Cytoscape[68] (version 3.9.1) We used logistic regression to model the cassette score distribution in macrophage clusters between adult shoulder capsule and RA synovial tissues, which was statistically evaluated using Wald test.

**Comparison of macrophages from frozen shoulder and knee synovium.** Data (Seurat Object) for macrophages from the knee synovium was a gift from the corresponding authors of #PMID:32601335[24] and can be retrieved from ArrayExpress under the accession E-MTAB-8322. We retained 19078 genes that had matching Ensembl ID in both datasets. We projected the annotations of the reference data, i.e., the adult shoulder capsule on to the cells from the knee synovium through semi-supervised machine learning using scANVI of scArches[26] with condition as the batch variable. The reference data and query data were both trained for 400 epochs. We accepted the predicted label of each query cell based on the highest score for the cluster (we observed minimal differences when a threshold of 0.6 or 0.8 was applied, data not shown).

**Analysis of single-cell RNA-seq data from foetal tissues.** Single cell RNA sequencing data from foetal tissues were analysed using the same workflow as described in the adult atlas analysis section except that use of an integration algorithm was found not to be necessary. The foetal data comprised of n = 17718 cells, of which 4538 cells were removed for abnormal gene count and high mitochondria gene expression (threshold as before). We selected n = 6000 HVG and utilised the top n = 16 PCs for computation of the UMAP and nearest neighbour graph. After removing a further n = 519 cells that had a low

gene count), low UMI count (threshold as before), HBB+ cells, we retained 12,661 high-quality cells. The count matrices for each of the cell-type regions were extracted and separately re-analysed. For the stromal cells we used n = 2000 HVG, n = 14 PCs and a clustering resolution of 0.7. For the myeloid cells we used n = 500 HVG, n = 11 PCs and a clustering resolution of 0.6. For T cells n = 1000 HVG, n = 7 PCs and a clustering resolution of 0.2 was used. For B cells n = 500 HVG, n = 7 PCs and a clustering resolution of 0.3 was used. The markers of each cluster were identified using a Wilcox test (BH adjusted P-value ≤ 0.05).

### Computational analysis of bulk RNA-sequencing data

**Data processing and differential expression analysis.** We assessed the quality of the paired-end bulk RNA sequencing data using fastQC and generated count matrix using release 32 of the Gencode transcripts reference panel (Grch38.p13) with Salmon[69](version 1.5.2). We performed dimension reduction (plotPCA; DESeq2) to determine the validity of the sample labels, and the sources of variation, and to visualize the structure of the data after the batch effect (stimulation) was adjusted using Limma[70] (removeBatchEffect version 3.50.0). Only genes that were expressed (counts≥10) in all the samples in any of the condition groups were kept for analysis (n = 17544 genes retained). Differential expression analysis was performed using DESeq2[62] (version 1.34.0 apeglm shrinkage), and the statistical significance between the full model (~sample + stimulation) and the reduced model (~sample) was evaluated using LRT.

**Cell type deconvolution analysis.** Deconvolution of the bulk RNA sequencing data to the adult single cell sequencing data was performed using MuSiC[71] (version 0.2.0). We used expression dataset objects from the raw count matrix of the single cell data using BisqueRNA[72] (version 1.0.5) and from the Bulk RNA sequencing data using Biobase[73](version 2.54.0). Differences in proportion of subtypes of cultured fibroblasts were evaluated for statistical significance using Kruskal-Wallis, followed by pairwise post-hoc test with Dunn's.

### Reporting summary
Further information on research design is available in the Nature Portfolio Reporting Summary linked to this article.

## Data availability
The single cell-RNAseq data generated in this study have been deposited in the ENA database under accession code ERP143358 (https://www.ebi.ac.uk/ena/browser/view/PRJEB58305). The bulk-RNAseq data generated in this study have been deposited in the ENA database under accession code ERP143359. Human reference genome GRCh38 (GENCODE v32 / Ensembl 98) was used in the alignment of the bulk and single cell transcriptomic data. Source data are provided with this paper.

## Code availability
The code used to perform quantitative analysis of Cell DIVE images for Figure S1E is available at Zenodo https://zenodo.org/record/8138743. A lightweight interactive visualisation tool for the adult shoulder capsule scRNAseq data is available on https://capsule.ndorms.ox.ac.uk.

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

## Acknowledgements

The human foetal material was provided by the Joint MRC/Wellcome Trust (grant# MR/R006237/1) Human Developmental Biology Resource (http://hdbr.org). We thank the Oxford Genomics Centre at the Wellcome Centre for Human Genetics (funded by Wellcome Trust grant reference 203141/Z/16/Z) for the generation and initial processing of sequencing data. Cell DIVE equipment in the Digital Pathology Omics Core was generously funded by the Kennedy Trust for Rheumatology Research. Research at the Nuffield Department of Orthopaedics, Rheumatology and Musculoskeletal Sciences, University of Oxford is supported through the National Institute for Health Research Oxford Musculoskeletal Biomedical Research Centre. The views expressed are those of the authors and not necessarily those of the National Health Service and the National Institute for Health Research of the Department of Health. We acknowledge the following funding sources:

Versus Arthritis Career Development Fellowship (22425), Medical Research Council MR/S035850/1 Human Immune Discovery Initiative (0006565 and 0008181) & the NIHR Biomedical Research Centre, Oxford, Dunhill Medical Trust RTF1906\121, Rosetrees Trust (PGL21/10048), Wellcome Trust (222426/Z/21/Z), Coordenação de Aperfeiçoamento de Pessoal de Nível Superior – Brasil (CAPES)

## Author contributions

SGD initiated and directed the project. SGD, MTHN and CDB designed experiments. SGD performed experiments with RB, HG, SD, JEA, PJ, MG, JC, CCM, IR, MA, DW, AB, AL and KP. MTHN, KJ and SNS analysed and interpreted computational bioinformatic data. SGD, RC, AB, DW, PJ, CPH, KP and PK analysed and interpreted histological data. MKS, LM, SA provided data to perform cross-tissue comparison of MERTK+ macrophages. DF, JR, DG, MC and AJC contributed to analysis of experiments. SGD, MTHN, SNS and CDB drafted the manuscript. ICECAP Consortium: SGD is ICECAP Principal Investigator, JR is ICECAP Chief Investigator, AR is S Tees Site Lead, SG is Oxford Site Lead, SG CL, AT, SC and PH assisted in the collection of patient shoulder capsule tissues for the study, KW is ICECAP Senior Research Nurse and Study Co-ordinator, BW, DB, LVE comprise the Oxford Research Nurse team, MA, JJ, NC comprise the S Tees Research Nurse team, LA and LK co-ordinated the study at Oxford & S Tees sites. All authors read and edited the manuscript.

## Competing interests

CDB and MC have founders shares in Mestag Therapeutics. There are no other competing interests to declare.

## Additional information

Michael T. H. Ng [1], Rowie Borst[1,7], Hamez Gacaferi [1,7], Sarah Davidson[1], Jessica E. Ackerman [1], Peter A. Johnson [1], Caio C. Machado[1,2], Ian Reekie [1], Moustafa Attar [1], Dylan Windell [1], Mariola Kurowska-Stolarska [3], Lucy MacDonald[3], Stefano Alivernini [4], Micon Garvilles[1], Kathrin Jansen [1], Ananya Bhalla[1], Angela Lee [1], James Charlesworth [1], Rajat Chowdhury[1], Paul Klenerman[1], Kate Powell [1], Carl-Philip Hackstein [1], ICECAP Consortium*, Dominic Furniss [1], Jonathan Rees [1], Derek Gilroy [5], Mark Coles [1], Andrew J. Carr [1], Stephen N. Sansom [1,7], Christopher D. Buckley [1,7] & Stephanie G. Dakin [1]

[1]University of Oxford, Oxford, UK. [2]University of Sao Paulo, Sao Paulo, Brazil. [3]Research into Inflammatory Arthritis Centre Versus Arthritis (RACE), University of Glasgow, Glasgow, UK. [4]Fondazione Policlinico Universitario Agostino Gemelli – IRCCS, Rome, Italy. [5]University College London, London, UK. [7]These authors contributed equally: Rowie Borst, Hamez Gacaferi, Stephen N. Sansom, Christopher D. Buckley. *A list of authors and their affiliations appears at the end of the paper. ✉e-mail: michael.ng@ndorms.ox.ac.uk; stephanie.dakin@ndorms.ox.ac.uk

## ICECAP Consortium

Stephanie G. Dakin [1], Jonathan Rees [1], Amar Rangan[1,6], Stephen Gwilym[1], Christopher Little[1], Andrew Titchener[1], Salma Chaudhury[1], Philip Holland[6], Kim Wheway[1], Bridget Watkins[1], Debra Beazley[1], Lois Vesty-Edwards[1], Louise Appleton[1], Marc Atkinson[6], Lucksy Kottam[6], Juliet James[6] & Natalie Clark[6]

[6]South Tees NHS Trust, Middlesborough, UK.

