## [Peer Review File · Nature Communications]

A single cell atlas of frozen shoulder capsule identifies features associated with inflammatory fibrosis resolutionEditorial Note: This manuscript has been previously reviewed at another journal that is not operating a transparent peer review scheme. This document only contains reviewer comments and rebuttal letters for versions considered at *Nature Communications*.

REVIEWERS' COMMENTS

Reviewer #1 (Remarks to the Author):

The authors have provided additional experiments and important clarifications that help support their conclusions. As such they have satisfactorily addressed my concerns. The FS and embryonic shoulder capsule scRNAseq datasets comprise an important resource, and the comparative analysis with RA datasets is instructive of the similarities between these diseases. Their findings regarding MERTK+ macrophages further strengthen and support the pro-resolving role of these cells. I believe this paper is interesting and suitable for publication in *Nature Communications*.

As a mediator, I am also assessing the authors' responses to the revision points by Reviewer #2, which I believe the authors have adequately addressed. In more detail:

1. Given the origin of the tissue and the adequate number of cells per cluster, as well as the validations presented by the authors, the total number of cells in the single-cell analysis should not be an issue. The value of a longitudinal study of the disease is clear and acknowledged by the authors. Nevertheless, the dataset is an important resource even in the absence of the component of time.
2. The authors provide quantifications of histological data along with higher magnification images to show clearly the populations identified by their single-cell transcriptomic analysis and have thus addressed the reviewer's concerns.
3. I agree with the authors that the use of a surrogate animal model may not be representative of the studied disease, which is further supported by the pro-fibrotic role of liver macrophage-specific MERTK.
- 4,5. The authors have satisfactorily clarified these comments.

Reviewer #2 (Remarks to the Author):

The resubmitted manuscript adequately addressed my previously raised concerns.